# Dual function of OmpM as outer membrane tether and nutrient uptake channel in diderm Firmicutes

Augustinas Silale [1], Yiling Zhu[1], Jerzy Witwinowski [2], Robert E. Smith[2], Kahlan E. Newman [3], Satya P. Bhamidimarri[1], Arnaud Baslé[1], Syma Khalid [4], Christophe Beloin [5] ✉, Simonetta Gribaldo [2] ✉ & Bert van den Berg [1] ✉

The outer membrane (OM) in diderm, or Gram-negative, bacteria must be tethered to peptidoglycan for mechanical stability and to maintain cell morphology. Most diderm phyla from the Terrabacteria group have recently been shown to lack well-characterised OM attachment systems, but instead have OmpM, which could represent an ancestral tethering system in bacteria. Here, we have determined the structure of the most abundant OmpM protein from *Veillonella parvula* (diderm Firmicutes) by single particle cryogenic electron microscopy. We also characterised the channel properties of the transmembrane β-barrel of OmpM and investigated the structure and PG-binding properties of its periplasmic stalk region. Our results show that OM tethering and nutrient acquisition are genetically linked in *V. parvula*, and probably other diderm Terrabacteria. This dual function of OmpM may have played a role in the loss of the OM in ancestral bacteria and the emergence of monoderm bacterial lineages.

The defining feature of diderm, or Gram-negative, bacteria is the presence of an outer membrane (OM)[1,2]. This complex cell envelope component is an asymmetric lipid bilayer, usually containing lipopolysaccharide (LPS) in the outer leaflet and phospholipids in the inner leaflet. The first formal definition of an OM was subsequent to an electron microscopy analysis of the cell envelope of the diderm Firmicute *Veillonella parvula* (Negativicutes class)[3]. However, most of our knowledge about the OM derives from studies of Proteobacteria, especially *Escherichia coli*. While the OM provides mechanical stabilisation to the cell[4], diderm bacteria must tether the OM to peptidoglycan (PG) for additional mechanical stability and to maintain cell morphology. *E. coli* has three OM tethering systems: Braun's lipoprotein (Lpp) that covalently links the PG with the OM[5,6], the lipoprotein Pal and the OM protein OmpA which associate with PG noncovalently[7–11]. Mutants of these systems exhibit cell envelope defects

and increased susceptibility to cell envelope stressors, such as detergents[7,12–15].

Recent phylogenetic analysis of the distribution of OM attachment systems throughout the Bacteria Tree of Life has highlighted a striking bimodal distribution across the two major clades in which Bacteria are divided, the Terrabacteria (including both classical monoderms such as Bacilli and Clostridia, and diderms such as Cyanobacteria and Deinococcus-Thermus) and the Gracilicutes (including only diderms such as Proteobacteria and Bacteroidetes)[16]. Braun's lipoprotein was shown to be only present in a subset of Proteobacteria, and Pal restricted to the Gracilicutes, together with the Lol machinery for lipoprotein export to the OM. OmpA showed a patchy distribution, being absent in many diderms. In contrast, the OmpM protein is a fourth OM tethering system that appears to be widespread in diderm Terrabacteria but is completely absent in the Gracilicutes[16].

[1]Biosciences Institute, Faculty of Medical Sciences, Newcastle University, Framlington Place, NE2 4HH Newcastle upon Tyne, UK. [2]Institut Pasteur, Université de Paris Cité, Unit Evolutionary Biology of the Microbial Cell, Paris, France. [3]School of Chemistry, University of Southampton, Southampton SO17 1BJ, UK. [4]Department of Biochemistry, University of Oxford, Oxford OX1 3QU, UK. [5]Institut Pasteur, Université de Paris Cité, Genetics of Biofilms Laboratory, Paris, France. ✉e-mail: christophe.beloin@pasteur.fr; simonetta.gribaldo@pasteur.fr; bert.van-den-berg@newcastle.ac.uk

OmpM consists of an N-terminal periplasmic S-layer homology (SLH) domain connected via a linker region to a C-terminal OM β-barrel[17–20]. The SLH domain has been mostly studied for its role in the building of S-layers in monoderm bacteria[21]. The primary role of the SLH domain is binding to the cell wall. Similarly to the SLH domain found in monoderm S-layer proteins[22–24], the SLH domain of OmpM proteins of some diderm Terrabacteria phyla such as Deinococcus-Thermus and Cyanobacteria has been shown in vitro to bind pyruvylated secondary cell wall polymers (SCWPs)[25–27], whereas in members of the Negativicutes (including *V. parvula*) it anchors the OM to PG modified with polyamines (e.g. cadaverine and putrescine) on the D-glutamate of the peptide stem[18,28,29].

Deletion of three out of the four OmpM paralogues in *V. parvula* resulted in a dramatic phenotype where the OM detaches, the periplasmic space is greatly enlarged and multiple cells share a single OM[16]. Complementation by the most abundant *V. parvula* OmpM paralogue, OmpM1, reverted this phenotype back to wild type, demonstrating in vivo that OmpM is responsible for tethering the OM to PG[16]. This dramatic phenotype after deletion of OmpM-like proteins has been reported also in two other representatives of diderm Terrabacteria, *Thermus thermophilus*[30] and *Deinoccocus radiodurans*[31,32].

These results led to the proposal that OmpM proteins represent an ancestral OM tethering system in Terrabacteria and possibly the last bacterial common ancestor, which united anchoring and nutrient acquisition functions[16]. The presence of this unique OM tether in Terrabacteria might therefore have been involved in the multiple OM-loss events inferred to have occurred specifically in this clade, perhaps via OmpM mutations weakening (and eventually abolishing) the SLH-PG interactions[16].

Due to its LPS component, the OM presents a permeability barrier for hydrophilic and hydrophobic small molecules, both detrimental (e.g. antibiotics) and essential (e.g. nutrients)[33,34]. Controlled permeability of the OM is established by OM proteins, which either mediate energized transport of specific nutrients in the case of TonB-dependent transporters, or allow size-limited diffusion in the case of porins (<~600 Da for rigid spherical molecules[33,34], although larger flexible molecules can also go through[35]). Porins are well-characterised in many Proteobacteria, but much less information is available on small-molecule permeation in diderm Terrabacteria. Earlier work on the OmpM-like protein Mep45 from the Negativicute *Selenomonas ruminantium* suggested that this protein can transport nutrients via its β-barrel domain[19], but another study in *Synechocystis* (Cyanobacteria) claimed that its OmpM orthologues form channels too small for most simple nutrients[20]. In *V. parvula*, the role of OmpM in nutrient acquisition has not been studied. In addition to the four OmpM paralogues, there are two other putative porin genes in the *V. parvula* genome: an OmpA-like homologue (*FNLLGLLA_00518*) and a porin (*FNLLGLLA_00833*)[36]. Proteomic analysis of the *V. parvula* OM showed that OmpM1, OmpM2 and FNLLGLLA_00833 are, respectively, the first, third, and sixth most abundant OM proteins[36], suggesting that these proteins could play major roles in determining OM permeability.

A recent analysis has proposed that the barrels of most OmpM-like proteins have 16–30 β-strands[32], and could therefore form pores that are large enough for small molecule diffusion across the OM, a hypothesis supported by electrophysiology experiments with *Deinococcus radiodurans* SlpA protein[37] and liposome swelling studies on Mep45 from *S. ruminantium*[19]. The recently determined structures of SlpA revealed that it forms a large trimeric complex composed of 30-stranded β-barrels, i.e. much larger than those of the general *E. coli* porins OmpC and OmpF[32,37,38]. However, since *D. radiodurans* has an unusually complex cell envelope with an S-layer[31,39,40], it is unclear how generalisable these findings are to OmpM proteins in other diderm Terrabacteria, especially in Negativicutes which tether the OM to polyaminated PG rather than pyruvylated SCWP.

Here we use single particle cryo-EM, X-ray crystallography, molecular dynamics simulations, bioinformatic analyses and functional assays to show that OmpM1 from *V. parvula* (VpOmpM1) is a general porin with similar structural and functional properties to *E. coli* OmpF, with an additional mobile periplasmic region that can bind *V. parvula* PG and take on different folds. We also show that the four OmpM paralogues are likely the only porins in *V. parvula*, implying that nutrient acquisition and OM attachment are genetically linked in diderm Firmicutes, and likely all diderm Terrabacteria, via OmpM. Our results support an ancestral dual function of OmpM as both an OM tether and nutrient uptake porin which may have supported life in early bacteria.

## Results

### VpOmpM1 is a trimeric porin with an extended periplasmic region

We determined the structure of VpOmpM1 expressed in *E. coli* by single-particle cryo-EM (Fig. 1a–d and Supplementary Table 1). Reconstructions up to 3.2 Å without enforced symmetry revealed a trimeric arrangement of the protomers. The C-terminal 16-stranded β-barrels that reside in the OM form a classical three-fold symmetrical porin assembly reminiscent of the well-characterised *E. coli* porins OmpF (EcOmpF) and OmpC[41,42]. However, the β-barrel lumens are constricted by inward-folded extracellular loops 3 and 7 in VpOmpM1 (Fig. 1b), rather than solely by loop 3 as in the *E. coli* porins. The inter-protomer contact area, mediated by hydrophobic sidechains on the outside of the β-barrel, completely excludes lipid/detergent and solvent. The β-barrels exhibit other features common to known porin structures, including an aromatic girdle at the membrane-solvent interfaces and strong density surrounding the β-barrels that likely corresponds to the lipid moieties of LPS and phospholipid or detergent (Fig. 1b, d and Supplementary Fig. 1). An unusual feature observed in the cryo-EM reconstructions was a weaker, elongated density that corresponds to the N-terminal SLH domain at the distal part, connected to the β-barrel via a triple coiled-coil (Fig. 1c, d). We term this periplasmic region of VpOmpM1 the 'stalk'. Only the region of the stalk proximal to the OM was resolved well enough for model building, as the cryo-EM density deteriorates towards the SLH domain end of the stalk. We speculated that the poor density is the result of movement of the stalk relative to the β-barrels. Measurements of the blurry SLH domain density observed in 2D class averages suggest that the distal part of the stalk spans 50–60 Å (Fig. 1c). The VpOmpM1 trimer is pseudosymmetrical: while the β-barrels exhibit three-fold symmetry, the stalk is tilted relative to the membrane plane normal (Fig. 1d). The pseudosymmetry was confirmed by enforcing C3 symmetry during data processing, which resulted in a better-resolved β-barrel region (2.8 Å) but uninterpretable density for the periplasmic stalk (Supplementary Figs. 2–4).

The *E. coli* cell wall does not contain polyaminated PG, the native ligand of VpOmpM1. Therefore, to exclude the possibility that in our structure of VpOmpM1 purified from *E. coli* the stalk region is misfolded due to the absence of polyaminated PG, we purified a His-tagged OmpM1 construct expressed in *V. parvula* (native VpOmpM1) and determined its structure by single particle cryo-EM to 3.3 Å resolution (Supplementary Table 1, Supplementary Figs. 2–4). The β-barrel structures are virtually identical (Cα–Cα root-mean-square deviation 0.239 Å), but there are differences in the relative orientation of the stalk (Fig. 1e). In reconstructions without enforced symmetry, the α-helices forming the coiled-coil are not equivalent in either dataset, as they exit the OM plane in different orientations (Fig. 1f). The reasons for these structural differences are not obvious. We speculate that intra- or inter-protomer hydrogen bonds at the stalk-barrel interface could be responsible for the subtle conformational differences. In the *E. coli* VpOmpM1 structure, the sidechain of R113 in one protomer interacts with the sidechain of N155 (part of the periplasmic end of the

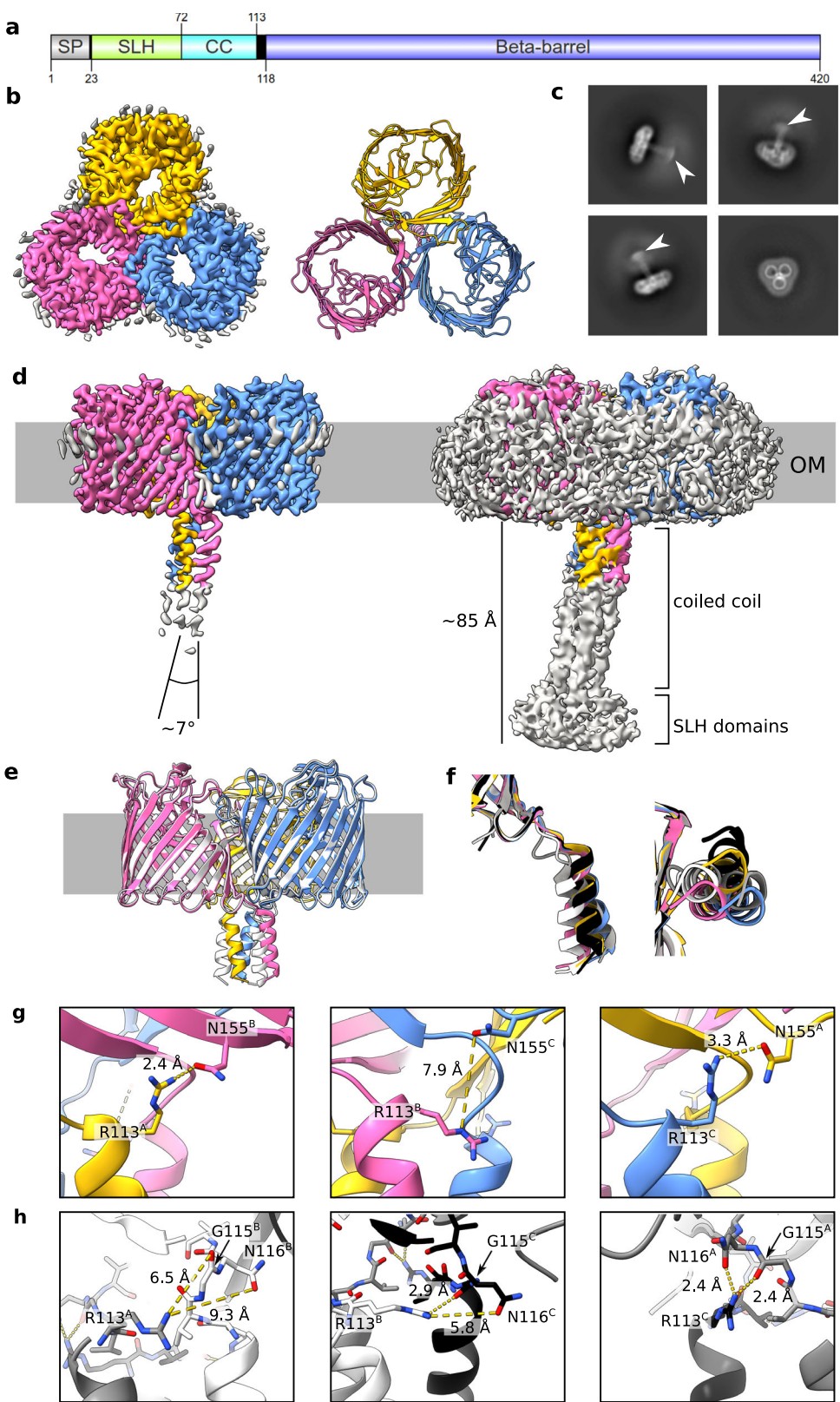

β2 strand) of another protomer. This interaction is observed only in two inter-protomer interfaces, but not the third one (Fig. 1g). A similar interaction was observed in the native VpOmpM1: at two inter-protomer interfaces, R113 interacts with the carbonyl oxygen of G115 and the sidechain of N116 (both part of the β-barrel end of the stalk) (Fig. 1h). Interestingly, the R113-N155 interaction was not observed in the native VpOmpM1 structure, and the R113-G115/N116

interaction was not observed in the VpOmpM1 structure from *E. coli*. It is possible that the two protein constructs differ slightly in their lowest energy states which are observed in the averaged cryo-EM data.

We performed all-atom molecular dynamics (MD) simulations to investigate the conformational space sampled by the stalk. We simulated the native VpOmpM1 model to which an AlphaFold2[43] prediction of the unmodeled region of the stalk was grafted and fit into the

**Fig. 1 | Cryo-EM structures of VpOmpM1 expressed in *E. coli* and *V. parvula*.**
**a** Schematic depicting VpOmpM1 (UniProt A0A100YN03) domain arrangement and boundaries. SP, signal peptide; SLH, S-layer homology; CC, coiled-coil. Generated using IBS 2.0[103]. **b** Structure of recombinant VpOmpM1. The cryo-EM density is shown on the left and the model is shown on the right as viewed towards the OM from outside the cell. **c** Representative 2D class averages. White arrowheads point to the diffuse SLH domain density. The edge of each square is 344.4 Å long. **d** Side view of the cryo-EM density at high (left) and low (right) contour level. The grey bar represents the outer membrane (OM). The diffuse grey density around the

membrane region in the high contour view likely corresponds to lipid or detergent. **e** Superposition of structures from recombinant VpOmpM1 (in colour) and native VpOmpM1 purified from *V. parvula* (white). **f** Superposition of coiled-coils of protomers from the recombinant (yellow, pink, blue) and native (grey, white, black) VpOmpM1 structures. The structural alignment was performed on the β-barrel region only (not shown). Left−side view, right−view from the periplasm towards the OM. **g**, **h** Inter-protomer contacts at the β-barrel-stalk interface in the recombinant (**g**) and native (**h**) structures. The protomer (A, B, C) for each sidechain is indicated in superscript.

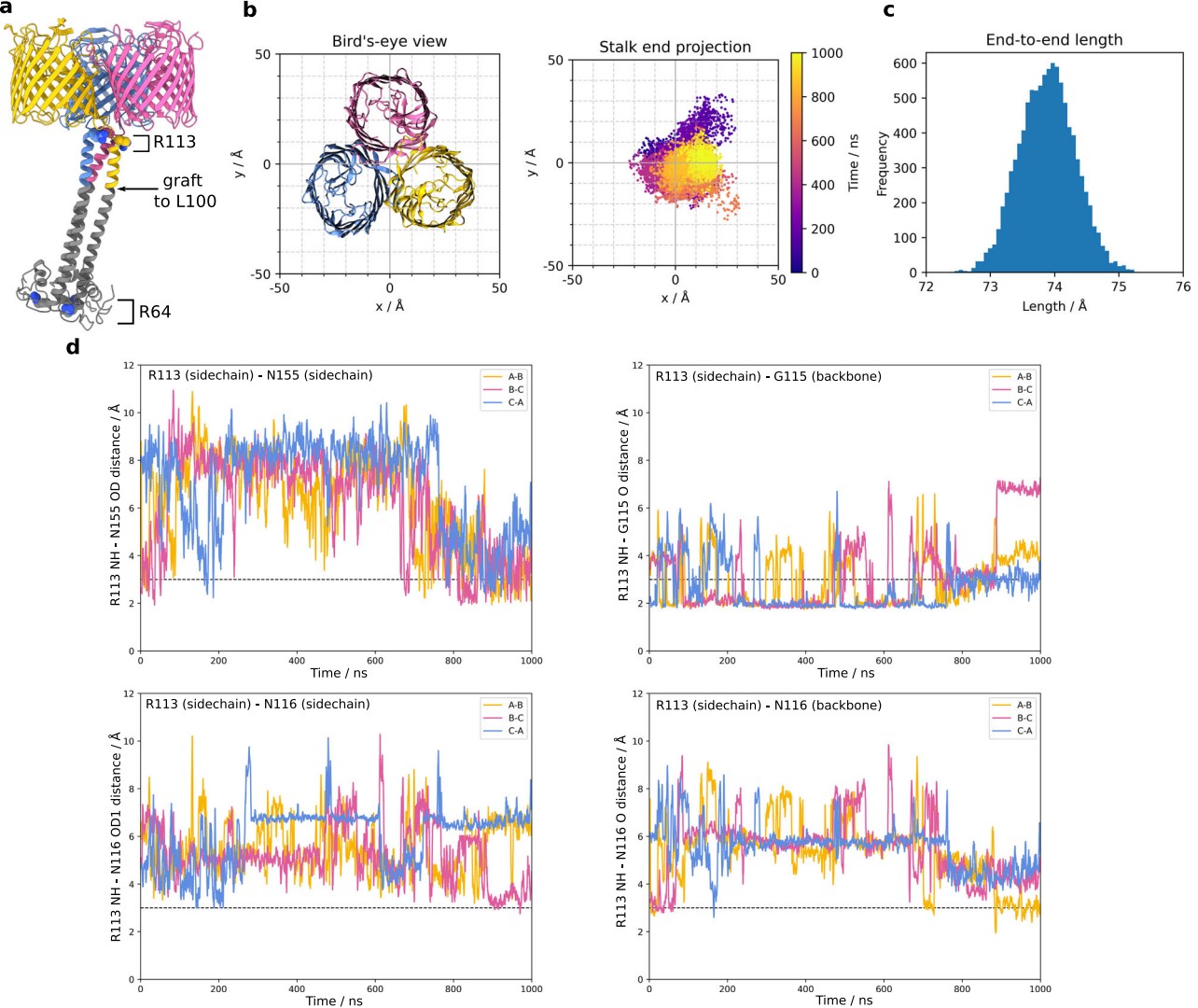

**Fig. 2 | All-atom molecular dynamics simulations with native VpOmpM1. a** The AlphaFold2 prediction for the stalk (grey) was grafted onto the N-terminus of the first residue (L100) modelled into the experimental density of native VpOmpM1 (in colour). R64 (bottom of SLH domain) and R113 are shown as space filling models. **b** Bird's-eye-view of the β-barrel region from outside the OM (left) and a plot of the stalk end projection (centre of mass of the R64 Cα atoms) over the 1 μs simulation

(right). The observation point in the left and right panels is equivalent. **c** Frequency distribution of stalk lengths (R64-R113 Cα-Cα) observed throughout the simulation. **d** Distances between the R113 sidechain and interacting sidechain atoms throughout the simulation. The dotted line denotes an inter-atomic distance of 3 Å. The equivalent plots for a replicate simulation and a simulation without the AlphaFold2 graft are presented in Supplementary Fig. 5.

low-resolution cryo-EM density (Fig. 2a). The simulations revealed substantial movement of the stalk in the periplasmic space (Supplementary Movie 1). The SLH domain end of the stalk sampled a span of ~50 Å (Fig. 2b) during the simulations, in agreement with the cryo-EM data (Fig. 1c). The coiled-coil does not kink throughout the 1 μs simulation (Fig. 2c), and the movement of the stalk is due to flexibility at the interface between the stalk and the β-barrels. We observed that the inter-protomer hydrogen bonds identified in our static cryo-EM

structures, R113-N155 and R113-G115, are broken and re-formed throughout the simulation (Fig. 2d). The R113 interaction with the sidechain of N116 was not stable, and R113 instead interacted with the backbone carbonyl oxygen of N116 (Fig. 2d). Similar results were obtained from a replicate simulation with the AlphaFold2 graft model as well as the native VpOmpM1 cryo-EM model by itself (Supplementary Fig. 5). The stalk would presumably be bound to the relatively immobile PG in vivo, which would significantly impair its mobility

relative to the OM. We speculate that the flexible interface between the stalk and the β-barrels would result in the ability of the OM to move slightly relative to the PG, imparting mechanical resistance to the cell envelope.

## The stalk binds polyaminated peptidoglycan

The SLH domain of OmpM binds polyaminated PG in vivo, as demonstrated in the Negativicute *S. ruminantium*[18,28]. Recent evidence for VpOmpM1 binding to PG comes from fluorescence microscopy and cryogenic electron tomography experiments that showed OM detachment when three *ompM* paralogues were deleted[16]. We tested binding of recombinant full-length and barrel-only VpOmpM1 to sacculi isolated from *V. parvula* and *E. coli* (Fig. 3a). Only full-length protein binds to *V. parvula* sacculi, as expected. Minor amounts of full-length VpOmpM1 were found in the wash fraction, suggesting that the non-covalent PG-SLH domain interaction is reasonably strong. No binding was observed to sacculi from *E. coli*, confirming that PG stem peptide polyamination is required for binding.

The cryo-EM density for VpOmpM1 SLH domains was not sufficiently resolved for model building due to movement of the stalk, as demonstrated by MD simulations. We therefore used bioinformatics and structural comparisons to investigate where polyaminated PG could bind within the AlphaFold2 predicted model of VpOmpM1 SLH domains. Monoderm bacteria attach cell surface proteins to SCWPs using single-chain proteins that have triple tandem SLH domain repeats, which fold into a core three α-helix bundle with additional helices and loops packing against this bundle[22,44–47]. Available crystal structures of the SLH domains 1–3 of *Bacillus anthracis* surface array protein (Sap) and *Paenibacillus alvei* S-layer protein SpaA are similar to the predicted VpOmpM1 SLH domain[44,45] (Fig. 3b) with Cα-Cα r.m.s.d. values of 1.0 and 1.3 Å, respectively. The main difference is that both Sap and SpaA have short core helical bundles, but the VpOmpM1 central bundle extends into the long coiled-coil that connects it to the β-barrels. Structures of SpaA bound to defined SCWP ligands[45] show that the sugars bind in grooves between SLH repeats (Fig. 3b) and interact with conserved motifs therein: conserved tryptophan and glycine residues bind the sugar moiety (W34 and G58 in VpOmpM1), and the arginine of the TRAE motif forms a salt bridge with the ketal-pyruvate modification of the SCWP (residues 63–66, TRYE, in VpOmpM1). It is not surprising that these motifs are conserved in most OmpM-like proteins from diderm Terrabacteria because they are thought to bind pyruvylated SCWP, but not in the Negativicutes (Fig. 3c).

Negativicute OmpM proteins are thought to recognise the polyamine modification on the α-carboxylate of the D-glutamate in the PG peptide stem. Crystal structures of proteins bound to putrescine[48–50] and cadaverine[51,52] indicate that aromatic residues are involved in binding these polyamines via stacking interactions with the aliphatic chain of the polyamine. We next investigated if the SLH domains are different between OmpM and non-OmpM proteins (Fig. 3c). We found that the tyrosine residue Y36 is highly conserved in the OmpM SLH domains from the Negativicutes and slightly less in the other two lineages of diderm Firmicutes (Limnochordia and Halanaerobiales), but it is not conserved in the SLH domains of OmpM homologues from other diderm Terrabacteria as well as non-OmpM SLH domains (Fig. 3c, d). Also, Y65 of the TRYE motif is more conserved in the OmpM SLH domain of Negativicutes than in the other groups of sequences.

Mapping these residues onto the predicted VpOmpM1 SLH domain structure reveals that the conserved tyrosines are located away from the grooves between the SLH chains that contain the putative PG disaccharide binding site (Fig. 3e). We suggest that the peptide stem containing the polyamine moiety specifically in Negativicutes OmpM could extend away from the disaccharide binding groove to interact with either of the conserved tyrosines. Alternatively, or perhaps additionally, PG interaction could induce conformational changes that bring the tyrosine side chains closer to the binding groove.

## The stalk domain of OmpM may exist in compact and extended conformations

We generated a soluble construct encompassing almost the entire VpOmpM1 stalk region (residues 22–107) in *E. coli* and determined its crystal structure to 1.7 Å (Fig. 4a, Supplementary Table 2, PDB 8BZ2) in an attempt to verify the AlphaFold2 prediction for this region. Surprisingly, the crystallized stalk does not fold into a coiled-coil but forms a compact trimer composed mainly of α-helices (residues 23–105). The protomer interfaces form grooves on one side of the trimer that could potentially accommodate ligands (Fig. 4b). However, mapping of the previously identified putative PG-binding residues onto the stalk crystal structure shows they are spread across the structure (Supplementary Fig. 6), suggesting that this conformation does not bind PG. The residues that form the coiled-coil in the extended AlphaFold2 prediction instead form helical hairpins and extensive intra-protomer interactions rather than interact with the other chains in the asymmetric unit (Fig. 4c). We obtained a very similar structure, albeit at a lower resolution, with a longer construct that encompasses the entire stalk region (residues 22–118). A DALI[53] search of the PDB using the SLH model (PDB 8BZ2) showed that it has low structural similarity to other proteins in the database (Supplementary Fig. 5). The highest ranking hit was a blood group antigen binding adhesin (PDB 5F7L[54]) with a 5.7 Z-score and 2.5 Å Cα-Cα RMSD.

The protomer fold of the stalk observed in the crystal structures aligns well with an alternative, compact conformation predicted by AlphaFold2 (Fig. 4d, e, Supplementary Fig. 8). This was surprising as the confidence for this prediction was low, yet it agreed with the experimental structure. However, the trimerization interface in the compact stalk prediction is different: residues that are far away in the crystal structure (e.g., M75) interact in the predicted compact conformation (Fig. 3e). We considered that this compact state could be an unstable high energy conformation (hence the low prediction confidence), and that the stalk construct is stabilized in a similar conformation in the crystallization condition. Notably, the soluble SLH domain did not bind to *V. parvula* or *E. coli* sacculi (Fig. 4f), unlike the full-length VpOmpM1 (Fig. 3a). This suggests that the structure of the isolated SLH domain is different than in the full-length protein, as observed by cryo-EM. Therefore, we do not believe that the conformation observed in the crystal structure is the result of crystal packing.

All-atom MD simulation of VpOmpM1 with the compact stalk showed the stalk helix of one protomer reaching towards the OM, partially occluding the β-barrel of another protomer and forming salt bridges with its periplasmic turns (Fig. 4g and Supplementary Movie 2). Additionally, this conformation was stabilised by inter-protomer hydrogen bonds in the stalk region (Supplementary Fig. 9). The role of this interaction is unclear, but this simulation supports the dynamic nature of the stalk.

The two different stalk conformations of VpOmpM1 imply that either there is conformational switching under the appropriate conditions (e.g., cell envelope stress, presence of specific ligands), or that the protein commits to a particular conformation during biogenesis. In one replicate of the all-atom MD simulations of the native VpOmpM1-AlphaFold2 graft model (Fig. 2a) we observed unfolding of the very N-terminus of the SLH domain of one chain, which then proceeded to interact with residues of the coiled-coil (Supplementary Movie 3, Supplementary Fig. 10). This suggests some flexibility for the SLH domain, and perhaps in vivo this flexibility is important for the SLH domain to find its PG ligand. The coiled-coil, however, remained stable during the simulation. Protein melting temperature analysis by dynamic scanning calorimetry suggested that the melting

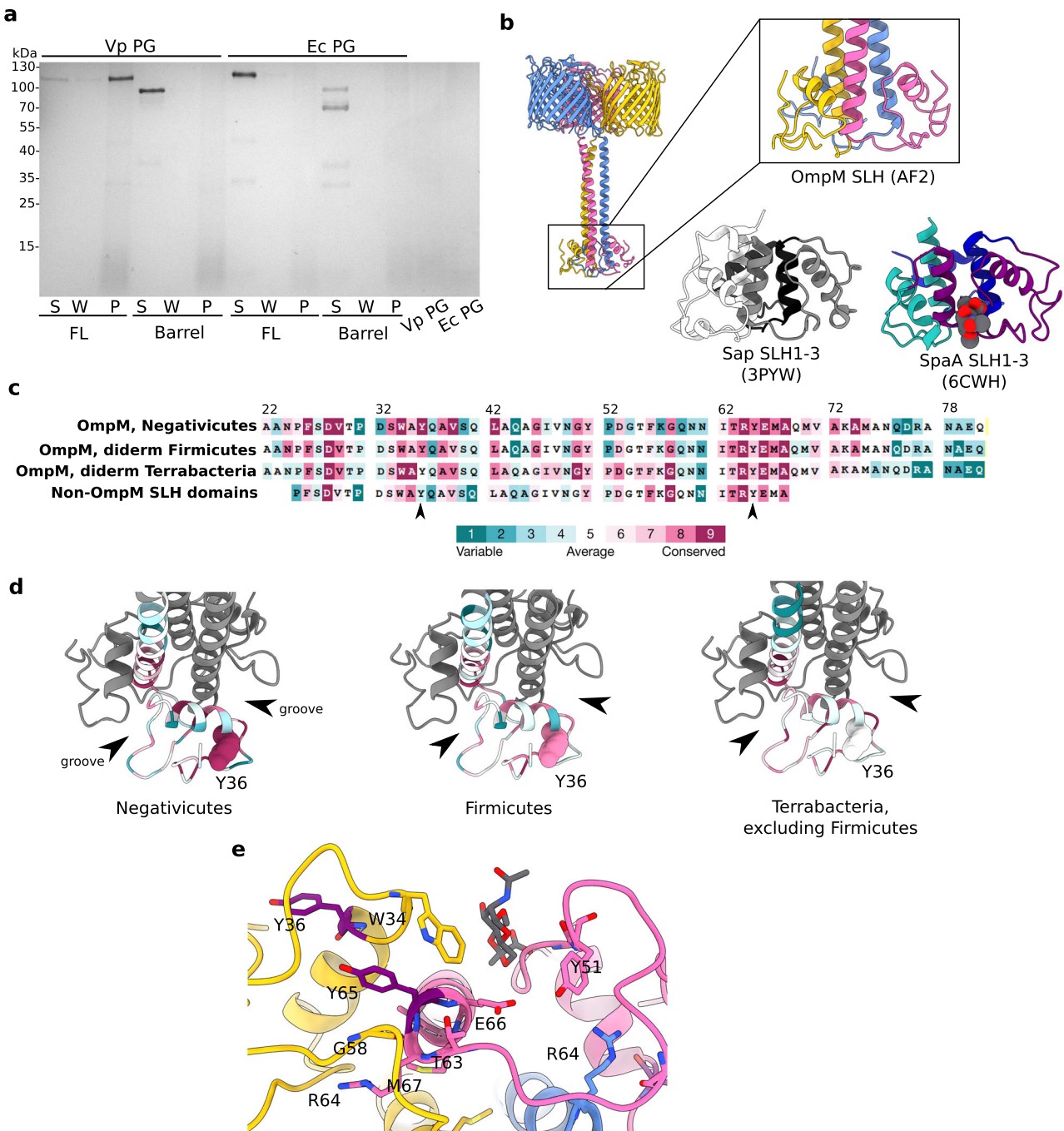

**Fig. 3 | Analysis of the putative peptidoglycan binding site within the SLH domain. a** Binding assay of recombinant full-length and barrel-only VpOmpM1 to sacculi (PG) isolated from *V. parvula* and *E. coli*. After incubation of protein with PG, the insoluble PG was pelleted by centrifugation, washed by resuspending in buffer, and pelleted again. The supernatant from the binding reaction (S), the wash (W) and the resuspended PG pellet (P) were boiled and analysed by SDS-PAGE. Three independent binding experiments were performed and yielded similar results. **b** Close-up view of the VparOmpM1 SLH domains predicted by AlphaFold2, the crystal structure of SLH domains 1–3 from *Bacillus anthracis* Sap (surface array protein) (PDB 3PYW)[44], and the crystal structure of SLH domains 1–3 from *Paenibacillus alvei* SpaA in complex with the monosaccharide 4,6-pyr-β-D-ManNAcOMe (space-filling representation) (PDB 6CWH)[45]. Views were generated from a superposition. **c** Multiple sequence alignment results from different subsets of SLH domain-containing proteins mapped onto the sequence of VpOmM1 SLH domain and coloured by conservation using ConSurf[104] (colour key). Row 1—OmpM homologues from Negativicutes; row 2—representative OmpM homologues from all diderm Firmicutes; row 3—representative OmpM homologues from diderm Terrabacteria, excluding Firmicutes; row 4—SLH domains from non-OmpM from diderm Firmicutes. The arrows point to Y36 and Y65, which could be important for binding polyaminated PG. **d** ConSurf results mapped onto the predicted SLH domains of VpOmpM1 (same sequence subsets and colour key as **b**). Arrows point to the grooves between SLH protomers. Y36 is shown in space-filling representation. The datasets used for ConSurf analysis correspond to the datasets used in a previous study[16] that were subsampled using custom scripts. **e** Conserved residues shown as stick models on the predicted SLH structure, coloured by chain, except for Y36 and Y65 which are in purple. SpaA bound to monosaccharide (PDB 6CWH) was superposed onto the VpOmpM1 SLH structure; the monosaccharide is in grey, the SpaA protein model is not shown.

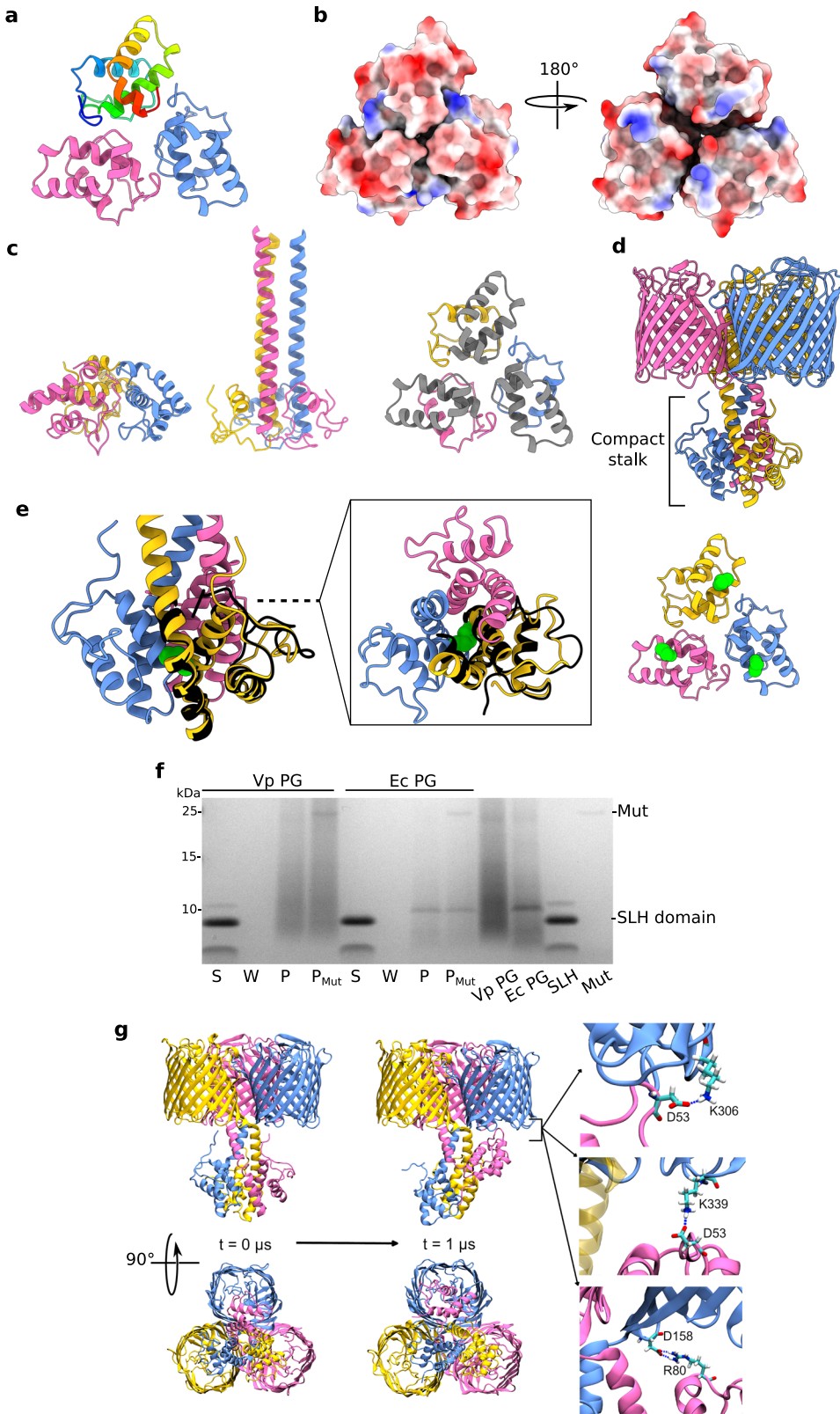

temperature of the stalk in the full-length VpOmpM1 construct is -87 °C (Supplementary Fig. 11), which means that the coiled-coil interaction is very stable and it is unlikely to unfold under physiological conditions. Therefore, the more favourable hypothesis is that the VpOmpM1 trimer is committed to either the extended or compact stalk conformation during biogenesis.

## OmpM paralogues are the only general diffusion porins in *V. parvula*

The β-barrel of VpOmpM1 consists of 16 anti-parallel β-strands connected by extracellular loops and periplasmic turns similar to the *E. coli* porin OmpF (Fig. 1b, c). Although sequence similarity of VpOmpM1 to *Ec*OmpF is only 25%, the overall structure is similar, but with some

**Fig. 4 | Crystallography, AlphaFold2, and binding assay show evidence of alternate stalk conformations. a** Crystal structure of VpOmpM1 stalk trimer at 1.7 Å. Residues 23–105 were resolved. One chain is in rainbow: N-terminus is blue, C-terminus is red. **b** Electrostatic surface presentation of the stalk crystal structure. (Left) Putative view from the OM and (right) from the periplasm. **c** Comparison of the stalk crystal structure (left) and AlphaFold2 predicted extended conformation (middle). Only residues 23–105 are displayed for both. (Right) Residues 64–105 of the stalk crystal structure that form the extended coiled-coil in the AlphaFold2 prediction are in grey. **d** Alternative, compact stalk conformation predicted by AlphaFold2. **e** Close-up view of the compact stalk prediction with a single chain from the crystal structure (black) superposed (Cα-Cα r.m.s.d. 0.78 Å) (left). (Middle) Top view down the section marked by the dashed line. (Right) Stalk crystal structure. The sidechain of M75 is shown in green space-filling representation in

each panel. **f** Binding assay of recombinant VpOmpM1 SLH domain to sacculi (PG) isolated from *V. parvula* and *E. coli*. After incubation of protein with PG, the insoluble PG was pelleted by centrifugation, washed by resuspending in buffer, and pelleted again. The PG was then resuspended and split into two aliquots, one of which was incubated with mutanolysin (Mut). The supernatant from the binding reaction (S), the wash (W) and the resuspended PG pellet without (P) and with Mut treatment ($P_{Mut}$) were boiled and analysed by SDS-PAGE. Three independent binding experiments were performed and yielded similar results. **g** All-atom MD simulation of AlphaFold2 predicted compact stalk model over 1 μs. The SLH domain of one of the protomers (pink) reaches towards the OM and interacts with the periplasmic turns of the β-barrel of another protomer (blue). The salt bridges shown on the right had occupancies of 5.8%, 3.44% and 1.89% (top to bottom) throughout the simulation.

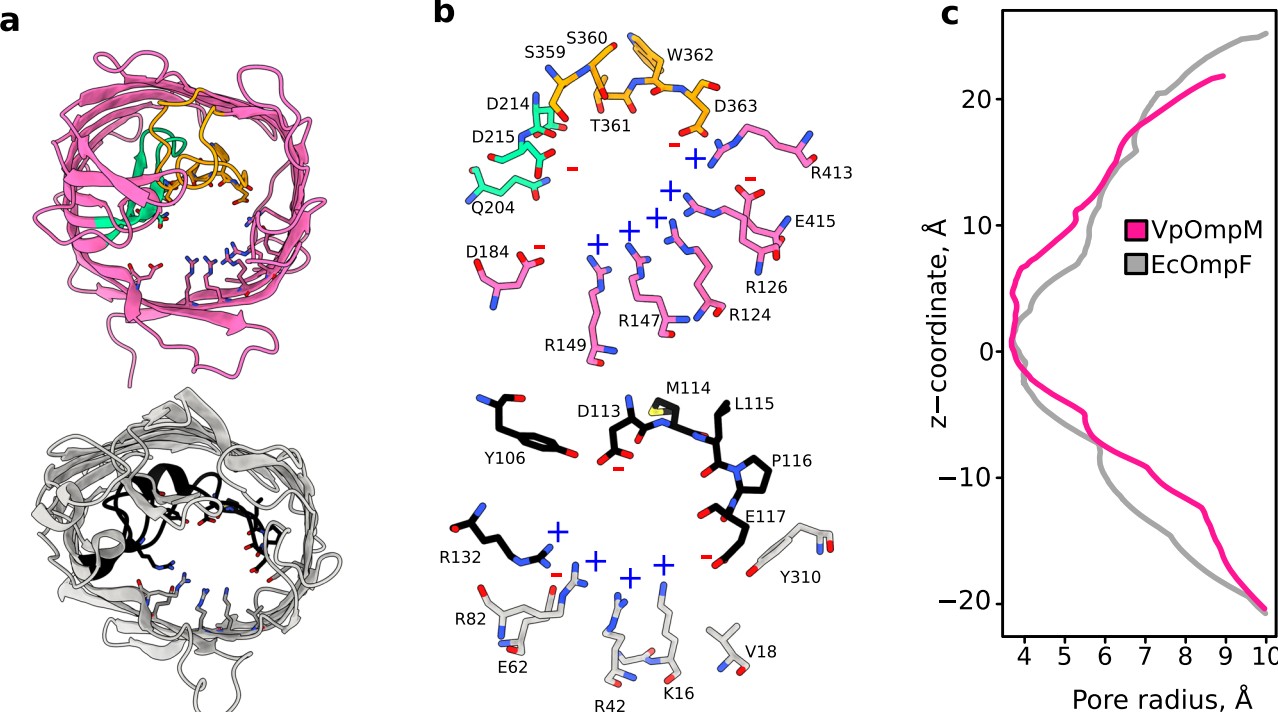

**Fig. 5 | VpOmpM1 and EcOmpF eyelet comparison. a** β-barrel from the C3 cryo-EM reconstruction of VpOmpM1 (top) and an EcOmpF protomer (PDB 3POQ)[105] (bottom). The views are generated from a superposition. VpOmpM1 loops 3 and 7 are in green and orange, respectively; EcOmpF loop 3 is in black. Residues forming the eyelet are shown in stick representation. **b** Close-up view of the eyelet regions of

VpOmpM1 (top) and EcOmpF (bottom). Colours as in (**a**). Positively and negatively charged residues lining the eyelet are annotated with a blue plus or red minus sign, respectively. **c** HOLE[55] profiles calculated from aligned experimental structures of VpOmpM1 β-barrel and EcOmpF. The z-coordinate is perpendicular to the membrane plane and its origin is at the narrowest part of the VpOmpM1 eyelet.

prominent differences in the extracellular loops (Fig. 5a and Supplementary Fig. 12). VpOmpM1 extracellular loops 3 and 7 fold into the β-barrel to form the constriction. In EcOmpF and similar Proteobacterial general porins the constriction is formed solely by loop 3 (Fig. 5a). The eyelet (the narrowest region of the pore) is formed by sidechains projecting from the β-barrel strands into the lumen as well as sidechain and backbone atoms of the constricting loop residues (Fig. 5b). Analysis of the pore radius of experimental structure β-barrels by the HOLE program[55] reveals a maximal constriction of ~3.7 Å (Fig. 5c). However, in EcOmpF the extent of the constriction (± 0.1 Å) along the axis perpendicular to the membrane plane is ~2.4 Å, and the VpOmpM1 constriction extends for ~7 Å. All-atom MD simulations show that the pore constriction in VpOmpM1 is constant due to an extensive interaction network restricting movement of the eyelet loops 3 and 7 (Supplementary Fig. 13). In agreement with a previous report[56], we observed that this is not the case in EcOmpF, where fluctuations in loop 3 can completely close the pore.

An electric field exists across the eyelet of general porins as a result of asymmetric distribution of charged residues[57,58], and a similar charge asymmetry is seen in the eyelets of VpOmpM1 and EcOmpF (Fig. 5b). We calculated the average dipole moment of water molecules inside the channel during the MD simulations and used this as a proxy for comparing the strength of the transverse electric field inside the two porins. The largest average dipole moment magnitudes are observed in the constriction regions (Supplementary Fig. 14), and the direction of the dipole points from the acidic residues towards the basic residues of the eyelet. This analysis shows that the EcOmpF constriction has a stronger localised transverse electric field, while VpOmpM1 has a weaker and more diffuse transverse electric field.

We tested the ability of VpOmpM1 to transport small molecules across membranes in liposome swelling assays (Fig. 6a). Notably, VpOmpM1 could transport lactate, which is essential for *V. parvula* growth, and putrescine, which is used for modification of the PG peptide stem. We also observed transport of the amino acids

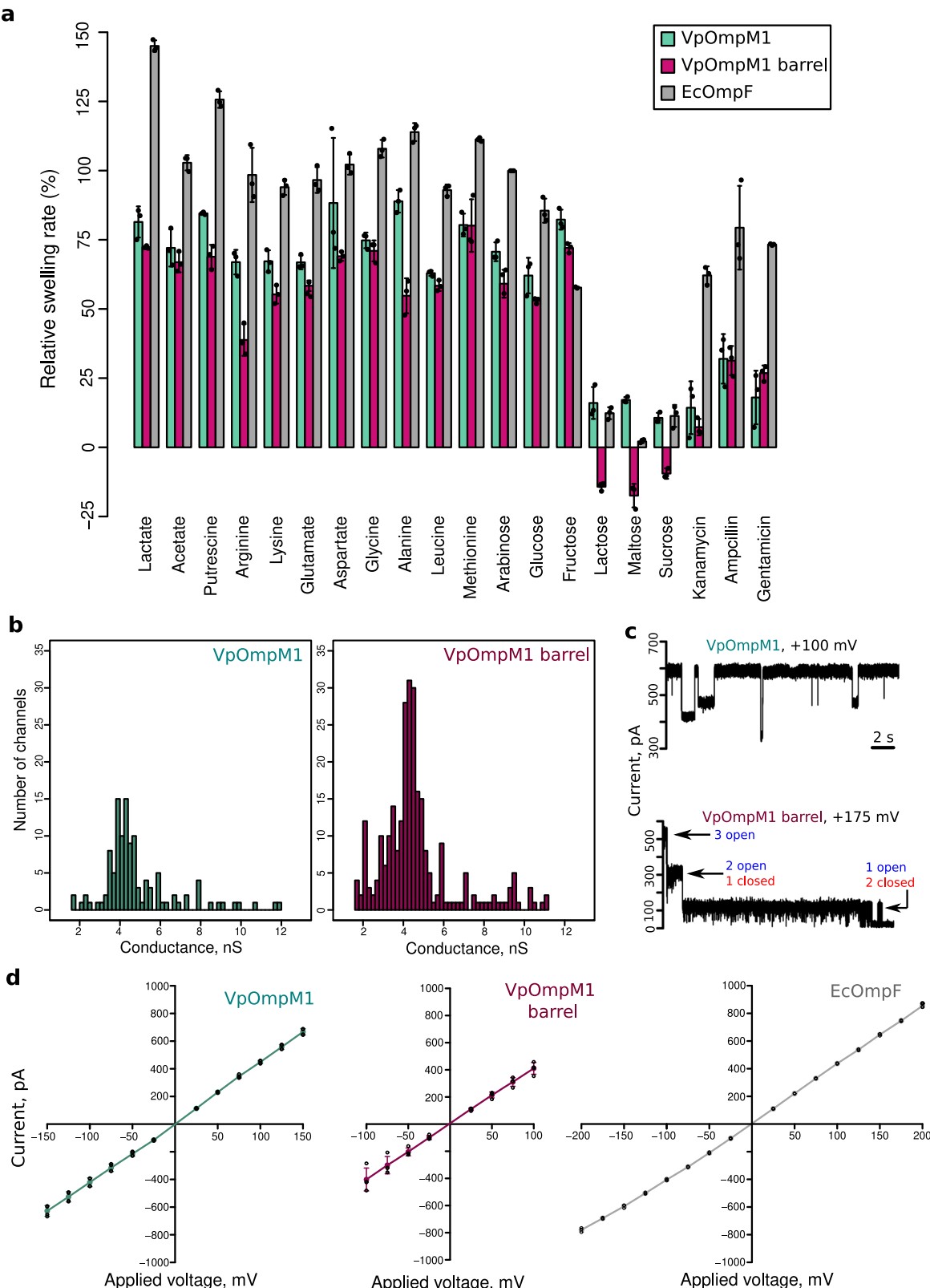

arginine, lysine, glutamate, aspartate, glycine, alanine, leucine, and methionine. VpOmpM1 transported monosaccharides, even though *V. parvula* is asaccharolytic, but not disaccharides, which underlines the non-specific, size-limited nature of transport. We carried out experiments with a VpOmpM1 construct that lacks the stalk to test if the stalk is involved in regulating transport, but this construct had similar transport properties to the full-length protein (Fig. 6a). The

only exception was impaired transport of arginine and alanine, and the reasons for this are unclear. VpOmpM1 had a similar substrate permeation profile to EcOmpF but showed consistently lower transport rates, which could be the consequence of a longer constriction region or weaker transverse electric field (Fig. 5c and Supplementary Fig. 14). One notable difference is that there was little transport of ampicillin, kanamycin and gentamicin by VpOmpM1

**Fig. 6 | VpOmpM1 channel properties. a** Liposome swelling in the presence of indicated substrate. Liposomes with embedded full-length VpOmpM1, VpOmpM1 barrel-only or EcOmpF were tested. Swelling rates were normalised to the rate of swelling of EcOmpF-containing liposomes in the presence of arabinose. Each condition was measured in technical triplicates (except EcOmpF with fructose, where $n = 2$) from the same proteoliposome preparation. Error bars show mean ± SD. **b** Bilayer electrophysiology conductance distribution plots obtained from recordings with multiple insertion events (see Supplementary Fig. 15 for representative traces). Full-length VpOmpM1 distribution is centred on 4.87 ± 1.86 nS (SD, $n = 117$); VpOmpM1 barrel-only distribution is centred on 4.66 ± 1.88 nS (SD, $n = 269$). **c** Representative bilayer electrophysiology recordings of full-length VpOmpM1 (top) showing multiple sub-conductance states and of the barrel-only construct (bottom) showing classical trimeric porin sequential channel closure at high voltage. **d** Current-voltage characteristics from single-channel recordings of full-length VpOmpM1 ($n = 4$), barrel-only VpOmpM1 ($n = 4$), and EcOmpF ($n = 3$). Individual data points are shown as open black circles and means as coloured dots. The error bars show the SD.

compared to EcOmpF (Fig. 6a), but the relevance of this for antibiotic resistance in vivo is unclear.

We further characterised the channel properties of VpOmpM1 in bilayer electrophysiology experiments. Initially, we obtained conductance value distributions from recordings with multiple channel insertion events (Fig. 6b and Supplementary Fig. 15a, b). The resulting distributions were very broad and centred at 4.87 ± 1.86 nS and 4.66 ± 1.88 nS for the full-length and barrel-only constructs of VpOmpM1, respectively. The broad conductance distributions are likely due to multiple sub-conductance states of the channel, as observed in recordings where the channels can randomly close to different extent (Fig. 6c and Supplementary Fig. 15c, d). We were able to get more accurate conductance values from single-channel recordings: 4.34 ± 0.46 nS (mean ± SD, $n = 8$) for full-length VpOmpM1 and 4.37 ± 0.28 nS (mean ± SD, $n = 8$) for barrel-only VpOmpM1. In single-channel recordings at higher voltages, we occasionally observed classical trimeric porin behaviour where the three protomer pores close sequentially (Fig. 6c and Supplementary Fig. 15c, d). Using our setup we obtained a conductance of 4.28 ± 0.19 (mean ± SD, $n = 9$) for EcOmpF, which is similar to previously reported values[59,60] and to VpOmpM1 conductance. Current-voltage characteristics for both VpOmpM1 constructs and EcOmpF were again very similar (Fig. 6d).

We conclude that VpOmpM1 is a general diffusion channel in *V. parvula*, with similar properties to EcOmpF. We expect that the VpOmpM2-4 paralogues also function as nutrient uptake channels based on their sequence similarity[16]. We think it is unlikely that the other two putative porins of *V. parvula*, FNLLGLLA_00518 and FNLLGLLA_00833, also transport nutrients because they are predicted to only have 10 strands in their β-barrels (Fig. 7a). Previous work on 10-stranded β-barrels has shown that their lumen is occluded by amino acid residue sidechains and that they cannot perform major transport roles (Fig. 7b)[61,62]. To verify this experimentally, we produced the β-barrel region of FNLLGLLA_00518 in sufficient quantities for liposome swelling experiments. It did not transport lactate and arabinose, and could only transport putrescine and glycine at much slower rates than VpOmpM1 and EcOmpF (Fig. 7c), in agreement with reported transport properties of small OM β-barrels[63,64]. Therefore, FNLLGLLA_00518 is not a general porin. We were not able to test the transport activity of FNLLGLLA_00833 because the protein could not be expressed in *E. coli*, but we expect this 10 β-strand OM protein to have very low substrate permeation rates as well.

## Discussion

Together, our results show that OmpM has a dual function as an OM tethering system and a nutrient uptake channel. Our structures, simulations and functional data provide insight into how these key OM-related functions are linked in diderm Firmicutes and probably many other diderm Terrabacteria that possess OmpM-like attachment systems and lack the well-characterised tethers present in the Gracilicutes.

Interestingly, structural data and MD simulations show that the extended stalk of VpOmpM1 is highly mobile. The functional consequences of this mobility are not clear, and neither is it clear why it may be preferable to a rigid connection between the PG and OM. One

possibility is that the stalk samples the local environment to find polyaminated-PG. Although one would expect the polyamine modification to be present on most peptide stems, PG spatial organization is not well understood, and it is unclear what proportion of the modified peptide stems is easily accessible to the SLH domains of VpOmpM1. Another possibility is that the flexible interface between the stalk and the β-barrels imparts favourable mechanical properties to the cell envelope, as having a somewhat flexible PG-OM tether would allow the OM to deform when external mechanical forces are applied.

The combination of structural, functional and bioinformatics data suggests the possibility that the stalk exists in multiple conformations, despite the fact we did not observe any particle populations with a compact stalk in our cryo-EM datasets. Unravelling the roles of both potential states of the stalk is a challenging problem that will likely have to be resolved using in vivo studies. Based on structural comparisons with SCWP-binding SLH domains from monoderm bacteria and our PG binding assay results, we can conclude that the extended stalk state observed in cryo-EM reconstructions of VpOmpM1 is the PG-binding state.

The stalked OmpM-like porins represent a new variation of SLH domains. The three copies of the SLH domain from separate protein chains come together to form three likely identical ligand-binding sites, whereas previously characterised triple SLH domains are encoded in a single chain and their ligand-binding grooves have different affinities for SCWPs due to differences in binding motif residues in each SLH repeat[45,47,65,66]. The functional consequences of having equivalent or non-equivalent ligand-binding sites are not clear in the context of protein anchoring to the PG/SCWP.

SlpA from *D. radiodurans* is the only other OmpM-like protein for which structures have been determined[32,37,38]. SlpA, like VpOmpM1, plays an important role in maintaining cell envelope integrity[16,31]. VpOmpM1 is similar to SlpA in its overall architecture: both proteins are trimers and have an N-terminal SLH domain connected via a coiled-coil to a C-terminal β-barrel domain. However, the VpOmpM1 β-barrel trimer is approximately the size of a single SlpA β-barrel (Supplementary Fig. 16). Consequently, the pore inside the SlpA β-barrel, although constricted by extracellular loop insertions, is much larger than in VpOmpM1 and is ~14 Å in diameter at the narrowest point. VpOmM1 clearly imposes a size filter for diffusion across the OM (Figs. 5 and 6) and prevents leakage of large periplasmic contents from the cell. SlpA has been shown to transport nutrients[37], but it is unclear how escape of periplasmic material via the massive SlpA β-barrel pore is prevented in vivo. Other components of the complex and unusual cell envelope of *D. radiodurans* could partially obstruct the SlpA pore[39,40]. A superoxide dismutase (DR_0644) was observed bound at the top of the stalk and the β-barrel interface in SlpA structures[32,38], and there are no homologues of this protein coded in the *V. parvula* genome. The authors of one study noted that large (28–30 strand) β-barrels in OmpM-like proteins appear to be confined to the Deinococcus-Thermus phylum[32]. Thus, the unusual β-barrel structure of SlpA and the superoxide dismutase partner protein might be adaptations of this phylum to resist environmental extremes, such as oxidative stress, radiation, vacuum conditions, and high temperatures[67].

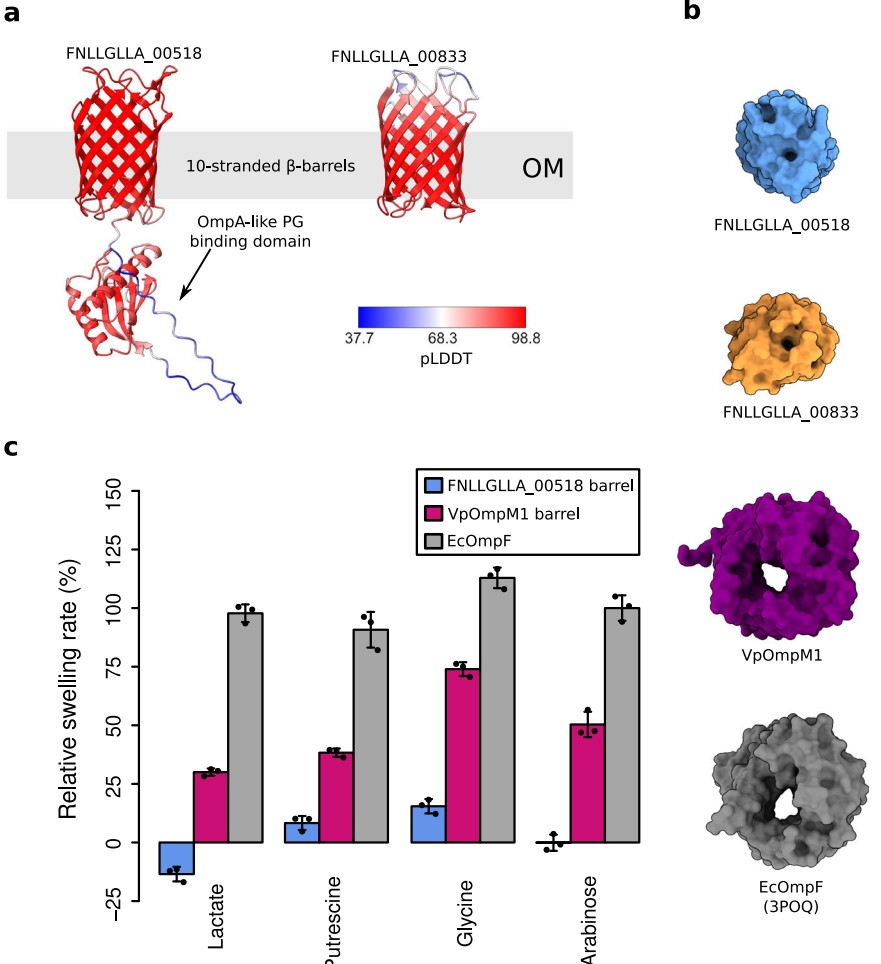

**Fig. 7 | *V. parvula* OM proteins other than OmpM paralogues are unlikely to be general diffusion channels. a** AlphaFold2[43] models of the FNLLGLLA_00518 (OmpA-like) and FNLLGLLA_00833 proteins. The colours represent the per residue confidence of the prediction (pLDDT). **b** Molecular surface models of *V. parvula* OM proteins and EcOmpF as viewed from outside the cell. **c** Comparative liposome swelling assays with FNLLGLLA_00518 β-barrel, VpOmpM1 β-barrel, and EcOmpF. Each condition was measured in technical triplicates from the same proteoliposome preparation. Bars show mean ± SD.

VpOmpM1, on the other hand, could be more representative of OmpM proteins in mesophilic diderm Terrabacteria. In fact, the β-barrel of Mep45, the OmpM homologue from the Negativicute *S. ruminantium*, has been shown to form a diffusion channel with similar properties to VpOmpM1, albeit with a larger estimated diameter (11.6 Å) than that observed for VpOmpM1 (7.4 Å)[19]. Cyanobacterial OmpM homologues have been shown to form small channels that facilitate transport of inorganic ions[20]. Our VpOmpM1 structures suggest that OmpM homologues from these and perhaps most diderm Terrabacteria might be similar to proteobacterial porins, unlike *D. radiodurans* SlpA. Our results indeed show that nutrient acquisition and OM tethering are genetically linked via OmpM in *V. parvula* and likely in all Negativicutes and other diderm Terrabacteria.

In conclusion, these data support the recent hypothesis that OmpM is an ancestral system that may have been present in the last common bacterial ancestor, possibly consolidating two key OM-related functions, tethering and nutrient uptake[16]. Therefore, deletion of or mutations in *ompM* may have promoted loss of the OM due to the concurrent loss of these fundamental cell functions. Although we envisage that OmpM plays similar roles in all diderm Terrabacteria, experimental characterisation of OmpM orthologues from diverse members of this clade will be required to fully capture the conservation and variation of this ancient OM tethering system.

## Methods

### Bacterial strains, culture conditions, and strain manipulation

Bacterial strains used in this work are listed in Supplementary Table 3. *E. coli* strains were genetically manipulated using standard laboratory procedures[68]. When needed, the following compounds were added to *E. coli* cultures at the following concentrations: ampicillin (liquid media) or ticarcillin (solid media)−100 mg/l, chloramphenicol−30 mg/l (liquid media) or 25 mg/l (solid media), kanamycin−50 mg/ml, apramycin−50 mg/l, diaminopimelic acid−300 μM.

*V. parvula* was manipulated as described previously[16,69,70]. When needed, the following compounds were added to *V. parvula* cultures at the following concentrations: chloramphenicol−25 mg/l, anhydrotetracycline−250 μg/l. The anaerobic conditions were generated using the GenBag Anaer generator (Biomérieux), or the GP Campus anaerobic chamber (Jacomex). The anaerobic chamber was filled with a H₂/CO₂/N₂ (5%/5%/90%) mixture.

### Primers, plasmids, and DNA manipulation

All plasmids and primers used in this study are listed in Supplementary Tables 4 and 5, respectively. Cloning was performed using either NEBuilder HiFi DNA Assembly Master Mix (New England Biolabs) or standard restriction cloning methods. Chemically competent *E. coli* DH5α or TOP10 cells[71] were used for transformation of cloning products or plasmids. *V. parvula* genomic DNA was extracted according to a protocol previously described for *Streptomyces* gDNA extraction[72]

from stationary phase cultures in SK medium[69]. PCR reactions for cloning applications were carried out using Phusion HiFi Master Mix (Thermo Fisher Scientific) according to manufacturer's protocol. PCR reactions for the control of constructs were carried out using the DreamTaq Green MasterMix (Thermo Fisher Scientific) or the EmeraldAmp GT PCR Master Mix (Takara Bio). Primers were obtained from Merck or Eurofins Genomics. PCR products were purified using the NucleoSpin Gel and PCR Clean-up kit (Macherey-Nagel). Restriction enzymes were of the FastDigest family of products (Thermo Fisher Scientific). Digestion products were isolated on agarose gels and purified with the NucleoSpin Gel and PCR Clean-up kit (Macherey-Nagel). Plasmid isolation was performed with NucleoSpin Plasmid kit (Macherey-Nagel). Sequence in silico manipulation was carried out using SnapGene (GSL Biotech, www.snapgene.com) and Geneious (Dotmatics). Primers were designed with NEBuilder (New England Biolabs, nebuilder.neb.com). Construct Sanger sequencing was performed by Eurofins (eurofinsgenomics.eu).

## Construction of expression vectors

For full-length VpOmpM1 expression in *E. coli* we inserted the VpOmpM1 CDS without the signal peptide into the pB22 vector (adding *E. coli* TamB signal peptide and seven histidine residues to the N-terminus). Briefly, we amplified the VpOmpM1 fragment with the JW203/JW202 primer pair using *V. parvula* SKV38 gDNA as a template and cloned it into pB22 digested with XhoI/XbaI, yielding the pJW46 vector. Similarly, the β-barrel portion of VpOmpM1 (pJW45) and FNLLGLLA_00518 (pB22-00518_21-200), and full-length FNLLGLLA_00833 (pB22-00833) were cloned into pB22. The *ompM1* region coding for the stalk (residues 22–107) was amplified from *V. parvula* SKV38 gDNA with the primer pair stalk_F/R, digested with NcoI and XhoI, and cloned into pET28b yielding a C-terminal His$_6$ fusion. For the expression of full-length VpOmpM1 in *V. parvula*, we inserted the C-terminal His-tagged *ompM1* coding gene containing the native ribosome binding site into SacI-digested pRPF185 using the JW172/JW206 primer pair and *V. parvula* SKV38 gDNA as a template, yielding vector pJW48. The vector was then transferred by conjugation into the Δ*ompM1-3 V. parvula* mutant strain as described previously[16].

## Protein expression and purification in *E. coli*

*E. coli* C43(DE3) Δ*cyoABCD* (Cyo complex deletion to improve purity) cells were transformed with a pB22 plasmid carrying either the full-length VpOmpM1 (pJW46), barrel-only VpOmpM1 (pJW45), barrel-only FNLLGLLA_00518 (pB22-00518_21-200) or full-length FNLLGLLA_00833 (pB22-00833). After overnight incubation at 37 °C, transformants were picked from Lysogeny Broth (LB)-ampicillin plates and used to inoculate a starter LB-ampicillin culture incubated at 37 °C with shaking for 2 h. Flasks with 1 l LB were inoculated with 8–12 ml of the starter culture and incubated at 37 °C, 160 rpm until OD600 ~ 0.5–0.6. Protein expression was induced by supplementing the cultures with 0.1% arabinose, followed by a further 3–4 h incubation at 37 °C with shaking. Cultures were harvested and cell pellets were stored at −20 °C. Cell pellets were thawed, resuspended in cold 20 mM Tris-HCl pH 8.0, 300 mM NaCl (TBS) and supplemented with DNase I. Cells were lysed by passing the cell suspension once through a cell disruptor (Constant Systems) at 23 kpsi. The lysate was clarified by centrifugation at $30,000 \times g$, 4 °C for 30 min. The membranes were isolated from the clarified lysate by ultracentrifugation at $140,000 \times g$, 4 °C for 50 min. The membranes were solubilised in 2.5% Elugent (Millipore) in TBS for 1 h at 4 °C. Insoluble material was pelleted by centrifugation at $44,000 \times g$, 4 °C for 30 min. The solubilised fraction was passed through a ~4 ml chelating sepharose column charged with Ni$^{2+}$ ions. The column was washed with 20 column volumes of TBS with 30 mM imidazole and 0.15% lauryldimethylamine oxide (LDAO), and bound protein was eluted with TBS supplemented with

200 mM imidazole and 0.2% decyl maltoside (DM). The eluate was concentrated using an Amicon Ultra filtration device (50 kDa cut-off membrane), loaded on a HiLoad Superdex 200 16/60 column and eluted in 10 mM HEPES-NaOH pH 7.5, 100 mM NaCl, 0.12% DM. Fractions were analysed by SDS-PAGE, pooled, concentrated by filtration (100 kDa cut-off membrane) and flash-frozen in liquid nitrogen. Protein samples were stored at −80 °C.

*E. coli* BL21 (DE3) cells were transformed with the pET28b-SLH_22-107 plasmid for production of the VpOmpM1 stalk region. Expression cultures were set up as above, except kanamycin instead of ampicillin was used in all media, and protein expression was induced by adding 0.4 mM isopropyl β-D-1-thiogalactopyranoside. Protein purification was performed as above, omitting the ultracentrifugation and solubilisation steps, and without detergents in buffers. A 10 kDa cut-off membrane was used for concentrating protein.

*E. coli* BL21 (DE3) cells were transformed with the pBAD24-EcOmpF plasmid for production of EcOmpF. Protein expression and purification was performed as for the pB22 constructs up to the membrane isolation stage. Given that pBAD24-EcOmpF doesn't encode a His-tag, inner membranes were selectively solubilised in 20 mM HEPES-NaOH pH 7.5 and 0.5% (w/v) sodium lauroyl sarcosinate for 30 min at room temperature with stirring. The insoluble fraction containing the outer membranes was recovered by ultracentrifugation for 30 min at $140,000 \times g$. The sarcosinate wash and ultracentrifugation steps were repeated once. Outer membranes were solubilised in 20 mM HEPES-NaOH pH 7.5, 50 mM NaCl and 1.5% LDAO for 1 h at 4 °C. The extract was clarified by ultracentrifugation and loaded on a 1 ml ResourceQ anion exchange column (Cytiva). The column was washed and eluted with a 0–500 mM NaCl gradient. Fractions containing EcOmpF were concentrated and subjected to size exclusion chromatography on a Superdex200 10/300 GL column (10 mM HEPES-NaOH pH 7.5, 100 mM NaCl, 0.05% LDAO). The protein was further purified on a MonoQ 5/50 GL anion exchange column (Cytiva), and detergent was exchanged to 0.12% DM via a final size exclusion chromatography run.

## Expression and purification of VpOmpM1 from *V. parvula*

15 l of *V. parvula* was grown overnight in anaerobic SK media containing 1.2% sodium lactate and supplemented with 700 μg/l sodium resazurin as an oxygen indicator. Briefly, media was mixed from sterile components, boiled to remove oxygen from the solution (until resazurin appeared colourless) and flushed for 15 min with N$_2$ to remove oxygen from the bottle head volume, before being sealed with a pierceable lid. Anaerobic bottles were inoculated from overnight starter cultures at an OD of 0.04 and supplemented with 25 mg/l chloramphenicol and 250 μg/l anhydrotetracycline. Cultures were grown at 37 °C overnight with 180 rpm agitation. Cells were harvested by centrifugation at $8000 \times g$ for 10 min. Pellets were pooled to represent 1.5 l of original culture each and resuspended in 5 ml HEPES pH 7.4. 100 μl $3.5 \times 10^4$ U/ml benzonase and a small spatula of lysozyme was added and sample was incubated on ice for 15 min. Cells were lysed by two rounds of French press and the debris was pelleted by centrifugation for 90 min at $15,000 \times g$ at 4 °C. The supernatant was carefully transferred to an ultracentrifuge tube containing 2 ml of a 50% sucrose cushion and centrifuged for 4 h at $35,000 \times g$. The volume at the sucrose supernatant interface was extracted, resuspended in 20 ml of HEPES pH 7.4 and pelleted at $35,000 \times g$ for 4 h. Pelleted membranes were frozen in liquid nitrogen and stored at −80 °C.

*V. parvula* membrane solubilization and the protein purification procedure was the same as for VpOmpM1 expressed in *E. coli*, except that the flow-through from the chelating sepharose column was put through the column again due to poor binding of protein of interest to the resin. The final size exclusion chromatography step was done on a Superdex 200 10/300 Increase column.

## Cryo-EM structure determination

Purified VpOmpM1 from *E. coli* (11.6 mg/ml) or *V. parvula* (8 mg/ml) was applied to glow discharged Quantifoil R 1.2/1.3 copper 200 mesh holey carbon grids. The grids were immediately blotted and plunge-frozen in liquid ethane using a Vitrobot Mark IV (ThermoFisher Scientific) device operating at 4 °C and 100% humidity. Data were collected on a FEI Glacios microscope operating at 200 kV using a Falcon 4 direct electron detector (ThermoFisher Scientific) at the University of York (Supplementary Table 1). A total of 4284 and 6505 movies with the *E. coli* and *V. parvula* samples, respectively, were recorded in electron event representation (EER) mode at 240,000 magnification, corresponding to a pixel size of 0.574 Å.

All image processing was done in cryoSPARC v3.3.2[73,74]. Data processing workflows are depicted in Supplementary Fig. 2. Movies were motion corrected using patch motion correction, and CTF parameters were estimated using patch CTF estimation. For VpOmpM1 expressed in *E. coli*, 2D classes generated from manually picked particles were used for template-based picking. 838,051 particles were extracted in 600 pixel boxes and Fourier cropped to 300 pixel boxes corresponding to a pixel size of 1.148 Å. Three rounds of 2D classification were used to discard bad particles, followed by generation of an ab initio 3D map using a stochastic gradient descent algorithm with 3 classes and 145,518 particles. All three classes were very similar, and a single class was used as a template in non-uniform refinement with the whole particle stack either without symmetry (C1) or with C3 symmetry enforced, with per-particle defocus and CTF group parameter (beam tilt and trefoil) refinement enabled. The 145,518 particle stack was subjected to heterogeneous refinements against the C1 map from non-uniform refinement and three decoy templates, or against the C3 map and the same three decoy templates. Particles were re-extracted in 600 pixel boxes with a pixel size of 0.574 Å. The final particle stacks (96,280 particles for the C1 map; 119,001 particles for the C3 map) were subjected to non-uniform refinement either with C1 or C3 symmetry enforced. The native VpOmpM1 dataset was processed similarly, except that two rounds of 2D classification were performed, four classes were used to make the ab initio map, four decoy templates were used in heterogeneous refinement, and the pixel size remained 1.148 Å throughout. Local resolution estimates, gold-standard Fourier shell correlations, and particle angular distribution plots for all maps are shown in Supplementary Fig. 3.

The final B-factor-sharpened maps from non-uniform refinement were used to build the models. A de novo model was built into the C3 map using Buccaneer[75] (part of CCPEM package[76]), followed by cycles of manual building in COOT[77] and real space refinement in Phenix[78]. The C3 model was then docked into the C1 map, manually extended in COOT and real-space-refined. The model built into the C1 map of VpOmpM1 expressed in *E. coli* was docked into the native VpOmpM1 C1 map, manually adjusted in COOT and real-space-refined. Models were validated using MolProbity[79]. Model refinement statistics are shown in Supplementary Table 1, and representative map-to-model fits are shown in Supplementary Fig. 4.

## Crystal structure determination

The purified VpOmpM1 stalk construct was concentrated to ~20 mg/ml. Sitting drop vapour diffusion crystallisation screens were set up using a Mosquito robot (SPT Labtech). Crystals grew in 0.1 M citric acid pH 3.5, 2.0 M ammonium sulphate at 20 °C. Crystals were cryo-protected in mother liquor supplemented with ~20% PEG400 and flash-cooled in liquid nitrogen. Diffraction data were collected at the synchrotron beamline I03 at Diamond Light Source (UK) at a temperature of 100 K (Supplementary Table 2). The dataset was processed with XIA2[80], scaled with Aimless[81], and the space group was confirmed with Pointless[82]. Data quality was evaluated in Xtriage[78]. Arcimboldo[83], part of the CCP4i2 suite[84], was used for ab initio phasing. The initial model was extended by Buccaneer and subjected to cycles of manual building and refinement in Phenix[78] (Supplementary Table 2).

## Isolation of sacculi

Peptidoglycan was isolated using the protocol of Wheeler et al.[85]. Briefly, cells were grown to exponential phase, harvested and boiled for 1 h in 4% SDS with agitation. The resulting insoluble sacculi were isolated by ultracentrifugation at 150,000 × *g* for 2 h. The pellet was then washed by 8 rounds of resuspension in milliQ H$_2$O and ultracentrifugation. Proteins were removed from the sample by incubating the cell pellet with 100 µg/ml trypsin in digestion buffer (50 mM Tris-HCl pH 7.0, 10 mM CaCl$_2$) overnight at 37 °C with agitation. To inactivate and remove the trypsin, sacculi were again boiled in 4% SDS and washed to remove residual SDS. Sacculi were then resuspended in MiliQ H$_2$O and lyophilized in preweighed Eppendorf tubes.

## Peptidoglycan binding assay

Dry *V. parvula* and *E. coli* sacculi were resuspended in binding assay buffer (10 mM HEPES-NaOH pH 7.5, 150 mM NaCl) at 20 mg/ml. Sacculi suspensions were sonicated for 10 min in an ultrasonic bath. Binding assays with full-length and barrel-only VpOmpM1 were carried out in 200 µl aliquots, each containing binding assay buffer with 1 mg/ml sacculus suspension, ~12 µg protein, and 0.12% DM. Binding reactions were incubated at 20 °C for 2 h. The sacculi were pelleted by centrifugation in a benchtop microfuge (21,000 × *g*) for 15 min at 8 °C. The supernatant was saved (the soluble fraction), the pellet was resuspended in binding buffer, and the sacculi were pelleted again. The second supernatant (the wash fraction) was saved, and the pellet was resuspended in binding buffer (the pellet fraction). Samples of the soluble, wash and pellet fractions were boiled in SDS loading buffer and analysed by SDS-PAGE on 12% FastCast gels (BioRad). The SLH domain binding assay was carried out with the following modifications. 36 µg of protein was used and the detergent was omitted. The resuspended pellet fraction was split into two 100 µl aliquots, and one aliquot was supplemented with 25 U mutanolysin (Sigma) and incubated at 37 °C for 30 min. The mutanolysin digestion was included to ensure that no SLH domain that could be bound to PG was prevented from entering the gel due to insolubility of the sacculi. The SLH binding assay was analysed by SDS-PAGE using pre-cast Bis-Tris 4–12% gels (Invitrogen) in MES running buffer.

## Computational modelling & simulation

**VpOmpM1 model building.** Computational models of trimeric full-length VpOmpM1 were predicted using AlphaFold2[43]. Two unique structures were predicted: one with an extended coiled-coil stalk, and one with a compacted stalk region (Supplementary Fig. 6). The AlphaFold2 predicted structure with the extended stalk displays no tilt of the stalk relative to the z-axis, in contrast to the reconstructed cryo-EM density. To generate a full-length VpOmpM1 structure with a tilted stalk consistent with the experimental data, the extended stalk of the AlphaFold2 prediction was fit into the low-resolution cryo-EM density and grafted to the well-resolved experimental structure using ChimeraX[86].

**System generation.** Three different models of VpOmpM1 were used to build protein-membrane systems: full-length VpOmpM1 with the grafted extended stalk; full-length VpOmpM1 with the compact stalk (AlphaFold2 predicted structure); and truncated VpOmpM1 from the reconstructed cryo-EM density (from residue L100 onwards). For each protein model, the β-barrel domain of the VpOmpM1 trimer was embedded in a model *Escherichia coli* outer membrane using the CHARMM-GUI Membrane Builder module[87]. The inner leaflet consisted of 90% 1-palmitoyl-2-oleoyl-sn-glycero-3-phosphoethanolamine (POPE), 5% 1-palmitoyl-2-oleoyl-sn-glycero-3-phosphoglycerol (POPG),

and 5% 1',3'-bis[1-palmitoyl-2-oleoyl-sn-glycero-3-phospho]glycerol (cardiolipin), and the outer leaflet consisted of 100% LPS (R1 core, 5 x O1 O-antigen units). This asymmetric bilayer system was solvated in 150 mM KCl, with additional calcium ions to neutralise the LPS headgroups (~115 mM in each system). The system composition is detailed in Supplementary Table 7.

**Atomistic molecular dynamics simulations.** All simulations used the CHARMM36m forcefield[88] with the TIP3P water model[89]. Simulations were carried out using the GROMACS simulation package (version 2021.2)[90]. For all simulation steps, a cut-off of 1.2 nm was applied to Lennard-Jones interactions and short-range electrostatics using the potential shift Verlet scheme. Long-range electrostatics were treated using the particle mesh-Ewald (PME) method[91]. Atoms were constrained using the LINCS algorithm to allow the use of a 2 fs timestep in NPT equilibration and production phases[92]. The three systems were energy minimised in 5000 steps using the steepest descent method[93]. The subsequent systems were equilibrated in six phases (two NVT and four NPT phases) in which the protein and lipid headgroups were subjected to position restraints with varying force constants (Supplementary Table 6). All equilibration phases used the Berendsen thermostat[94] to bring the system to 303.15 K (coupling constant of 1.0 ps). NPT equilibration phases used semi-isotropic Berendsen pressure coupling scheme[94] to equilibrate with a pressure bath of 1 bar ($\tau p = 5.0$ ps, compressibility of $4.5 \times 10^{-5}$ bar$^{-1}$).

These equilibrated systems were then simulated. The extended stalk system was simulated in duplicate for 1 μs, and the compacted and truncated stalk systems were each simulated as a single replicate for 1 μs. These production simulations utilised the Nosé-Hoover thermostat[95] (coupling constant of 1.0 ps) and the semi-isotropic Parrinello-Rahman barostat[96] ($\tau p = 5.0$ ps, compressibility of $4.5 \times 10^{-5}$ bar$^{-1}$). Trajectories were analysed using GROMACS tools and MDAnalysis utilities[97,98]. Water orientation was analysed using a locally-written python script (available with deposited data on Zenodo). Molecular graphics and supplementary animations were generated using VMD (version 1.9.4a51)[99] and Molywood (version 0.22)[100].

**OmpF simulation.** To compare the dynamics of the constriction regions, EcOmpF was also simulated. The crystal structure of trimeric EcOmpF (PDB ID: 2OMF[41]) was embedded in a model *E. coli* outer membrane and solvated using CHARMM-GUI, then equilibrated as described for the VpOmpM1 systems above. The system composition is shown in Supplementary Table 8. This system was simulated once for 500 ns under the same conditions as the above VpOmpM1 simulations.

**Bilayer electrophysiology**
Electrophysiology measurements were carried out in a custom-made cuvette at the centre of which was suspended a 25-μm-thick Teflon film (Goodfellow Cambridge Ltd) with a 80–100 μm aperture and 1.25 ml electrode chambers either side of the film. The Teflon film aperture was made lipophilic by painting with 1.5–3 μl 1% hexadecane in hexane solution on both sides, and the hexane was allowed to evaporate for at least 30 min. Electrophysiology buffer (10 mM HEPES-KOH pH 7.5, 1 M KCl) was added to both cuvette chambers, and 3 μl of 5 mg/ml 1,2-diphytanoyl-sn-glycero-3-phosphocholine (DPhPC) dissolved in *n*-pentane was added. The *n*-pentane was allowed to evaporate for at least 5 min. DPhPC bilayers were formed using the method of Montal and Mueller[101]. Pure, concentrated protein (7–15 mg/ml) was serially diluted in 1% Genapol X-100 (Sigma). 0.3–0.5 μl of diluted protein was added to the *cis* (ground side) chamber, and diluted with electrophysiology buffer as required to promote protein insertion into the bilayer. Protein insertion events were detected as sudden jumps in current in constant-voltage mode. All measurements were carried out using Ag/AgCl pellet electrodes attached to an Axopatch 200B amplifier headstage, and a Digidata 1440 A digitiser. The cuvette and headstage were enclosed in a custom-made Faraday shield during recording. Clampex software was used for recording. Clampfit software was used to analyse data.

**Liposome swelling assays**
Liposome swelling assays were carried out using the method of Nikaido and Rosenberg[102] with modifications. Lipid solution (4 mg/ml phosphatidylcholine and 0.46 mg/ml dihexadecyl phosphate in chloroform) was dried under an air stream, and completely dried under vacuum for at least 2 h. 80 μl of lipid solution was used per condition, i.e. per protein to be reconstituted. The dried lipids were resuspended in 100 μl deionised water per condition. Equimolar amounts of protein were added to the resuspended lipid so that the total protein quantity was 15–30 μg, followed by immediate vortexing. All conditions were supplemented with DM containing buffer to ensure that equal amounts of detergent were present in all samples. DM-containing buffer was added to control liposomes. Solutions were sonicated in a water bath for 1–1.5 min until translucent, and dried under vacuum overnight. The following day the proteoliposomes were rehydrated in 200 μl 10 mM HEPES-NaOH pH 7.5 and 12 mM stachyose per condition for 1–2 h at room temperature. Control liposomes were used to determine the concentration of each substrate that is isosmotic to the intraliposomal milieu (i.e. there is no swelling of the control liposomes), usually between 7.5 and 15 mM substrate in 10 mM HEPES-NaOH pH 7.5. Substrate concentrations used can be found in Supplementary Table 9. The decrease in $A_{400}$ due to liposome swelling was measured after adding 200 μl of the substrate solution to 15 μl of the proteoliposome suspension. Readings were taken on a Perkin Elmer Lambda 35 spectrophotometer in 1 s intervals for 30 s. A line was fit to the absorbance data corresponding to the linear phase of swelling (2–15 s), and the slope of the line was recorded as the swelling rate. Control liposome swelling rates were subtracted from all proteoliposome swelling rates. All data were normalized to EcOmpF-containing liposome swelling in the presence of arabinose.

**Reporting summary**
Further information on research design is available in the Nature Portfolio Reporting Summary linked to this article.

## Data availability

Electron microscopy maps have been deposited in the Electron Microscopy Data Bank with the accession codes EMD-16328 (VpOmpM1 from *E. coli* in C1), EMD-16333 (VpOmpM1 from *E. coli* in C3) and EMD-16332 (VpOmpM1 from *V. parvula* in C1). Atomic coordinates have been deposited in the Protein Data Bank under accession codes 8BYM (VpOmpM1 from *E. coli* in C1), 8BYT (VpOmpM1 from *E. coli* in C3), 8BYS (VpOmpM1 from *V. parvula* in C1), and 8BZ2 (VpOmpM1 stalk crystal structure). The following atomic coordinates used for comparison purposes were downloaded from the Protein Data Bank: 3PYW, 6CWH, 3POQ, 5F7L, 2OMF, 2ZFG, 8AGD. Trajectories and run input files for the Molecular Dynamics simulations are available on Zenodo (https://doi.org/10.5281/zenodo.8239075). Source data for electrophysiology, liposome swelling assays, and HOLE and MD analyses are available. Other data presented in this paper, constructs and strains are available on reasonable request. Source data are provided with this paper.

## Code availability

The water orientation analysis script is available on Zenodo (https://doi.org/10.5281/zenodo.8239075).

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

## Acknowledgements

We thank the University of York (UK) electron cryo-microscopy facility for access to instrumentation and support. We acknowledge the Diamond Light Source for I03 beamline access (proposal mx-24948) and support. We thank David Staunton (University of Oxford) for carrying out the dynamic scanning calorimetry experiments. This work was supported by a Wellcome Trust Investigator award (214222/Z/18/Z) to B.v.d.B., providing salary support to A.S. and Y.Z. C.B., S.G., J.W., and R.E.S. were supported by funding from the French National Research Agency (ANR) (no. Fir-OM ANR-16-CE12-0010), the Institut Pasteur Programmes Transversaux de Recherche (no. PTR 39-16), and the French government Investissement d'Avenir Program, Laboratoire d'Excellence Integrative Biology of Emerging Infectious Diseases (grant no. ANR-10-LABX-62-IBEID). The authors acknowledge the use of the IRIDIS High Performance Computing Facility, and associated support services at the University of Southampton, in the completion of this work. K.E.N. was supported by a Ph.D. Studentship from the Engineering and Physical Sciences Research Council (Project Number: 2446840), and S.K. by an EPSRC established Career Fellowship (EPSRC grant no. EP/V030779/1).

## Author contributions

A.S. made constructs, purified proteins, determined cryo-EM and crystal structures, carried out functional assays, and wrote the manuscript with input from all authors. Y.Z. purified proteins and carried out functional assays. J.W. made constructs and strains and prepared the dataset for ConSurf analysis. R.E.S. grew *V. parvula*, prepared membrane pellets, and isolated sacculi. K.E.N. performed simulations, supervised by S.K. S.P.B. collected preliminary electrophysiology data. A.B. collected X-ray diffraction data and managed the Newcastle Structural Biology Laboratory. B.v.d.B. made constructs, purified and crystallised proteins. S.G., C.B. and B.v.d.B. conceived and supervised the project.

## Competing interests

The authors declare no competing interests.
