## [Peer Review File · Nature Communications]

An ancestral dual function of OmpM as outer membrane tether and nutrient uptake channel in diderm FirmicutesReviewers' Comments:

Reviewer #1:

Remarks to the Author:

The manuscript by van den Berg and coworkers deals with the high-resolution structure of a new bacterial porin, OmpM, from the outer membrane (OM) of *Veillonella parvula* that shows interesting structural and dynamical properties. Contrarily to porins from *Enterobacter* species, the OmpM porin shows a long coiled-coil region that connects the β -barrel to the SLH domain, the stalk, which represents a tethering system to the peptidoglycan layer. This system is considered the ancient mechanism to provide mechanical stability to the OM, evolved later into other systems, such as the OmpA porin or the Braun's lipoprotein, both absent here. To note that recently the structure of a porin extracted from the S-layer of an extremophile bacterium (*Deinococcus Radiodurans*) was solved and showed a similar long coiled-coil system that probably connects the S-layer to the peptidoglycan.

The paper provides very interesting results in the field of bacteria evolution and structure of porins, based on robust data verified with several complementary techniques; it is well written and the content is for sure of broad interest, I read easily and with attentiveness.

Thus, I would recommend its publication after revision to the current version, with a few suggestions noted here:

- It is interesting the comparison with the most known bacterial porin, OmpF from *E. coli*. The authors reported the internal radius profile and the main residues of the constriction region, which appears longer than in OmpF. It would be possible to provide the alignment between OmpF and OmpM? What is the similarity among the two porins?
- The existence of an internal electric field in general bacterial porins (line 371-372) were stated with more recent papers than refs. 50-51. In addition, with the simulations the authors performed here, they should say something more precise about this property, providing a quantitative comparison with the electric field of OmpF.
- It is not clear to me what they analyzed with the software HOLE, the single high-resolution structure or the MD simulations? In principle from the trajectory of 1 μ s they have, it is possible to obtain more precise data about the size of the eyelet region (providing also the fluctuations), especially after some recent papers where OmpF has been described as very flexible, with the dynamics of the internal loop able to close completely the pore (Vasan, A. K. et al. *Proceedings of the National Academy of Sciences* 119, (2022).). Also, this extended analysis is easily doable, in light of an additional folded loop (L7) in OmpM.
- In the introduction (line 91) they still report the limit of 600 Dalton for passive transport of small-molecules through porins. This upper limit comes from an old paper from Nikaido. More recently a paper from Novartis researchers (Ruggiu, F. et al. *ACS Infect. Dis.* 5, 1688–1692 (2019).) stated that this limit is valid for spherical rigid molecules, thus with the possibility for larger and flexible molecules with a cylindrical shape to go through.
- Line 410: the conductance of OmpN is similar to OmpF, what about the selectivity?

Reviewer #2:

Remarks to the Author:

General Overview

Overall this is a very interesting work. The identification of another "stalked" porin, after the DR_2577 (SlpA) and its S-layer Deinoxanthin Binding Complex (SDBC) in *D. radiodurans*, suggests that this group of proteins could be more diffused than expected.

The reported findings are of primary importance for the field and in most of the parts results are exhaustively presented. However, some primary key points are underestimated or overlooked (see details below). Also, some part of the results sometimes seems uselessly extended with comments

that are evidently pertinence of the discussion section (see details below).

To my opinion the manuscript is worth for publication if the authors are willing to answer/amend/prove the aspects detailed in the following.

Major comments

Abstract section

The final part of the abstract should be rephrased to make it more readable for a broad audience. At the end of the abstract "...the multiple OM loss events that have been inferred in the Terrabacteria." is too implicit, please contextualize it and make it more explicit.

Introduction section

- Line 70, I would say some word more about the SLH domains in general

- Lines 109-112, in the citations a more recent (Jan 2023) structure of the SDBC (SlpA) has been published. It would benefit the manuscript to use that one because it is the only complete structure, including cofactors, stalk and SLH domain. In this last article (Jan 2023), the whole complex is modeled experimentally from the density maps (not by Alpha-fold.....)

- Lines 112-114, I would not cite a manuscript only showed on BioArchive (ref. 33), hence without peer review. There are instead two other papers that can be used as reference: one from the same authors of ref. 32 published on Nov 2021, and one from a Canadian group published on Dec 2021 and done with super-resolution microscopy

- Lines 125-126, unfortunately while a see a clear evidence for the nutrient uptake, the tethering function to me seems to be much less straightforward. What are the striking evidences for it?

Results section

- Lines 151-155, about the pseudo-symmetry, a local refinement approach can prove or disprove this statement which, at the moment, seems to be purely speculative considering the showed data. As reference, please consider the recently published structure for the SDBC (Jan 2023), which has a very extended and dynamic stalk. It should work also in your case.

What makes the authors excluding that the system has a very dynamic stalk that simply "oscillates" (with some preferential orientation on the grid) around the symmetry axis (as suggests figure 1d)? I think should be stated that this pseudo-symmetry may not be real in vivo.

-Lines 174-191, These considerations may be made on a wrong assumption (local refinement strategies must be applied). Is PG expected to bind proteins covalently? If yes, did you use some enzyme to cut the covalent bond allowing a rational isolation of the protein? If not, how can you explain a statical and stabilizing function in the presence of non covalent interactions?

In general, I would shorten this part; what makes a reference is what you observe in VpOmpM1 isolated from *V. parvula* (its own organisms) and there the stalk seems straight and not banded (Fig. 1e).

- Line 229, the second paragraph of results is an extended mix of results and discussion. I would shorten it by focusing as much as possible on the results.

- Lines 238-240, for comparison of SLH domains, you can also use the SDBC from *D. radiodurans* recently published. I am saying this because you also compared the OM region of your porin with the SDBC one (although using an incomplete structure).

Using the latest SDBC structure (published in Jan 2023) would benefit the manuscript because there you can find the complete model, including the entire stalk and SLH domain, therefore allowing you to

make better comparisons.

Lines 265-271, this is pertinence of the discussion section.

Line 295, is this a new paragraph? If yes, a title is missing. If not, it is not clear the connection with the lines above.

Lines 308-310, SLH domains are very conserved as sequence and structure. Could this mean that this is not an SLH domain but just a PG-binding domain?

Lines 318-326, again, this is pertinence of the discussion section

Figure 4, do you have any evidence of this in cryo-EM data? If this conformational changes are real, you should see the particles fall in different classes because the stalk compression is quite important. I am asking because in crystallography you might see "artifacts" due to the packing conditions and AlphaFold can only predict up to the knowledge it can find and at the moment might be too poor as there is only one published structure of a porin-like protein with a complete stalk and SLH (that in this manuscript is not even cited...)

Discussion section

- Line 464, I am not sure you can state so strongly the tethering function. In this respect, it seems to me that in vivo or in situ direct evidences are missing. Apart those indirectly inferred, what are these evidences? Can you shortly cite them?

It might be of help to do comparative experiments with WT strains and Mutants carrying OmpM without stalk. In this type of experiments, cryo-ET on intact cells may provide straight evidences proving or disproving your statements about tethering. I would expect from these experiments that something is loosen in the membranes. Alternatively, I would "tone down" this statement.

- Lines 477, as on the comment for Lines 308-310, SLH domains are very conserved as sequence and structure. Could this mean that this is not an SLH domain but just a PG-binding domain?

- Lines 482-484, Please, see comment for figure 4. To me it seems that if these multiple conformations would be the case, then cryo-EM would have surely revealed them. On the contrary, at the moment the cryo-EM suggests that this is unlikely. I would tone-down this statement.

- Lines 489-490, the SLH associated to the stalk was described and modeled previously on the SDBC in *D. radiodurans* (Jan 2023). Please, refer to this articles if somehow it was overlooked.

- Lines 511-512, The trimer of a β -sandwich protein (DR_0644) was observed around the stalk and in close proximity to the β -barrel interface in one SDBC structure. The DR_0644 protein was modeled in the last complete SDBC structure recently published and not mentioned in this manuscript (Jan 2023). The DR_0644 was found to be a superoxide dismutase (SOD), with several important biological implications. To me it seems that reference 29 is not the best option (it only shows an AlphaFold prediction and not functional/biological info).

Minor comments

Introduction section

- Lines 45-46 is written "...in presence of an additional, outer membrane", it seems that there is a second outer membrane. Maybe rephrase it as "...is the presence of a second external membrane, the Outer Membrane (OM)."

- Line 47 should be "lipopolysaccharides"

- Lines 51-53 I would rephrase this part. The mechanical stabilization of the OM on the cell is mainly possible because of the interaction with the PG. The PG does not provide additional stability but is (especially in the absence of an S-layer) a main static element of stability.
- On Lines 61 and 67, the term striking/ly to me seems redundant please use a synonym in one of the places.
- Line 74, maybe "or" should be replaced with "and"?
- Lines 81-85, I would split this sentence in two parts
- Line 97, nutrients
- Lines 99-100, I would write "a OmpA-like homologue (FNLLGLLA_00518) and a porin (FNLLGLLA_00833)"
- Line 194, "constructs" do you mean structures?

Results section

- Figure 1, Please respect the letter order, maybe you could merge (b) with (d)?
- Figure 1f (caption), Please specify that you are referring to the stalk region

Discussion section

- Lines 470-473, Is it known the distance between OM and IM in this species? Can the SLH domain be interacting with the inner membrane?
- Lines 473-480, Is this species carrying an S-layer? If yes, could it be also that the S-layer could help in providing rigidity making the system less disordered?
- Lines 498-499, another DR_2577 (SlpA) structure (into its complex, the SDBC) is currently available.
- Line 503, supplementary figure 12. Please, use the complete structure available for SDBC (Jan 2023), not the truncated version, at lower resolution and with poor integrity (e.g., no cofactors bound, no intact stalk, no SLH domain....)

Material and methods section

- Line 660, at which temperature the centrifugation? Why such a long time?

REVIEWER COMMENTS AND ANSWERS OF THE AUTHORS

Reviewer #1 (Remarks to the Author):

The manuscript by van den Berg and coworkers deals with the high-resolution structure of a new bacterial porin, OmpM, from the outer membrane (OM) of *Veillonella parvula* that shows interesting structural and dynamical properties. Contrarily to porins from *Enterobacter* species, the OmpM porin shows a long coiled-coil region that connect the b-barrel to the SLH domain, the stalk, which represent a tethering system to the peptidoglycan layer. This system is considered the ancient mechanism to provide mechanical stability to the OM, evolved later into other systems, such as the OmpA porin or the Braun's lipoprotein, both absent here. To note that recently the structure of a porin extracted from the S-layer of an extremophile bacterium (*Deinococcus Radiodurans*) was solved and showed a similar long coiled-coil system that probably connects the S-layer to the peptidoglycan.

The paper provides very interesting results in the field of bacteria evolution and structure of porins, based on robust data verified with several complementary techniques; it is well written and the content is for sure of broad interest, I read easily and with attentiveness.

Thus, I would recommend its publication after revision to the current version, with a few suggestions noted here:

- It is interesting the comparison with the most known bacterial porin, OmpF from *E.coli*. The authors reported the internal radius profile and the main residues of the constriction region, which appears longer than in OmpF. It would be possible to provide the alignment between OmpF and OmpM? What is the similarity among the two porins?

We thank the reviewer for the suggestion and have now included an alignment of EcOmpF and VpOmpM1 in panel a of Supplementary Figure 10. We note that the sequences are very different (8% identity, 25% similarity), and we had to align the EcOmpF and VpOmpM1 barrel structures using the RCSB PDB pairwise structure alignment tool in order to obtain a meaningful sequence alignment.

- The existence of an internal electric field in general bacterial porins (line 371-372) were stated with more recent papers than refs. 50-51. In addition, with the simulations the authors performed here, they should say something more precise about this property, providing a quantitative comparison with the electric field of OmpF.

We apologise for the omission. The references are now updated to more recent studies (PMID: 28485920, 26931352). In addition, we now include an analysis of the average orientation and magnitude for the sum of the dipole moments of water molecules inside the pore regions of VpOmpM1 and EcOmpF (Supplementary Figure 12). The data indicate that water molecules in both channels have preferred dipole moment orientations, indicating the presence of an electric field. The apparent transverse electric field is greater and more localised in EcOmpF, compared to the VpOmpM1 electric field which is less strong and more diffuse along the channel axis.

- it is not clear to me what they analyzed with the software HOLE, the single high-resolution structure or the MD simulations? In principle from the trajectory of 1 us they have, it is possible to obtain more precise data about the size of the eyelet region (providing also the fluctuations), especially after some recent papers where OmpF has been described as very flexible, with the dynamics of the internal loop able to close completely the pore (Vasan, A. K. et al. Proceedings of the National Academy of Sciences 119, (2022).). Also, this extended analysis is easily doable, in light of an additional folded loop (L7) in OmpM.

The HOLE plots shown in Figure 5c were calculated from the experimental structures, not from MD simulations. We thank the reviewer for the suggested analysis, which we now include in Supplementary Figure 11. We observed that the constriction loops L3 and L7 in VpOmpM1 are much less mobile than EcOmpF L3. This is likely due to stabilization of L3 and L7 by an extensive interaction network within and between the constriction loops. The EcOmpF results are consistent with the findings of the Vasan et al. study. Thus, based on our analysis, it seems that the VpOmpM1 constriction is much more rigid and does not fully close.

- In the introduction (line 91) they still report the limit of 600 Dalton for passive transport of small-molecules through porins. This upper limit comes from an old paper from Nikaido. More recently a paper from Novartis researchers (Ruggiu, F. et al. ACS Infect. Dis. 5, 1688–1692 (2019).) stated that this limit is valid for spherical rigid molecules, thus with the possibility for larger and flexible molecules with a cylindrical shape to go through.

We thank the reviewer for this comment. We now include the suggested reference and include changes to the text (lines 107-108).

- Line 410: the conductance of OmpN is similar to OmpF, what about the selectivity?

We thank the reviewer for this comment. We observed preferential translocation of potassium ions through the pore of both OmpM and OmpF in our equilibrium simulations. However, the number of translocation events was low (<70), so we cannot draw firm conclusions. Further simulations as well as electrophysiology measurements will be required to investigate the selectivity of OmpM. We believe the manuscript already has plenty of data and, therefore, investigation of selectivity is out of the scope of the paper.

Reviewer #2 (Remarks to the Author):

General Overview

Overall this is a very interesting work. The identification of another “stalked” porin, after the DR_2577 (SlpA) and its S-layer Deinnoxanthin Binding Complex (SDBC) in *D. radiodurans*, suggests that this group of proteins could be more diffused than expected.

The reported findings are of primary importance for the field and in most of the parts results are exhaustively presented. However, some primary key points are underestimated or overlooked (see details below). Also, some part of the results

sometimes seems uselessly extended with comments that are evidently pertinence of the discussion section (see details below).

To my opinion the manuscript is worth for publication if the authors are willing to answer/amend/prove the aspects detailed in the following.

Major comments

Abstract section

The final part of the abstract should be rephrased to make it more readable for a broad audience. At the end of the abstract “...the multiple OM loss events that have been inferred in the Terrabacteria.” is too implicit, please contextualize it and make it more explicit.

We thank the reviewer for pointing out the lack of clarity. We have rephrased this portion of the abstract (lines 45-48).

Introduction section

- Line 70, I would say some word more about the SLH domains in general

We have added some additional information about SLH domains in the introduction (lines 77-83).

- Lines 109-112, in the citations a more recent (Jan 2023) structure of the SDBC (SlpA) has been published. It would benefit the manuscript to use that one because it is the only complete structure, including cofactors, stalk and SLH domain. In this last article (Jan 2023), the whole complex is modeled experimentally from the density maps (not by Alpha-fold.....)

We thank the reviewer for pointing out this omission. We now reference the Farci et al. 2023 study (line 128, ref. 39) and include a structural comparison of full-length SDBC/SlpA and OmpM, as requested below (Supplementary Figure 14).

- Lines 112-114, I would not cite a manuscript only showed on BioArchive (ref. 33), hence without peer review. There are instead two other papers that can be used as reference: one from the same authors of ref. 32 published on Nov 2021, and one from a Canadian group published on Dec 2021 and done with super-resolution microscopy

The manuscript that the reviewer refers to has since been published in PNAS. We have updated the reference, and also included the references suggested by the reviewer.

- Lines 125-126, unfortunately while a see a clear evidence for the nutrient uptake, the tethering function to me seems to be much less straightforward. What are the striking evidences for it?

Thank you for your comment. In Witwinowski et al. 2022 (ref. 16; PMID: 35246664) the deletion of three OmpM paralogues in *V. parvula* results in OM detachment from

the PG, and the OM forms large vesicles containing multiple cells sharing a common periplasm. This deficiency can be completely reverted to the wild type phenotype by expression of OmpM1. Although this is indicative of OmpM1 performing an OM tethering function, we concur that OmpM homologues from *V. parvula* have never been shown biochemically to bind PG. We now include experiments where we tested binding to sacculi isolated from *V. parvula* and *E. coli* (negative control lacking polyamination on the peptide stem) by recombinant full-length OmpM1, OmpM1 barrel (Figure 3a) and SLH domain (Figure 4f). We find that only full-length OmpM1 binds to sacculi from *V. parvula* but not from *E. coli*. We consider these results to be conclusive evidence for the tethering function of OmpM proteins.

Results section

- Lines 151-155, about the pseudo-symmetry, a local refinement approach can prove or disprove this statement which, at the moment, seems to be purely speculative considering the showed data. As reference, please consider the recently published structure for the SDBC (Jan 2023), which has a very extended and dynamic stalk. It should work also in your case.

What makes the authors excluding that the system has a very dynamic stalk that simply “oscillates” (with some preferential orientation on the grid) around the symmetry axis (as suggests figure 1d)? I think should be stated that this pseudo-symmetry may not be real in vivo.

We made extensive efforts to resolve the structure of the stalk in cryo-EM reconstructions, including local refinement masking out the transmembrane portion, using particle subtraction to remove the signal from the transmembrane region, 3D variability analysis and 3D classification. However, we were unable to get reconstructions of the stalk that would allow model building. We believe this is because the stalk is only ~30 kDa in size, and focused refinement of such a small region invariably leads to overfitting noise and poor reconstructions. We note that the whole OmpM trimer is around 130 kDa (excluding the detergent micelle), and we managed to obtain 3.2 Å reconstructions using a 200 kV electron microscope. We do not think that collecting more or better (300 kV) data would allow structural determination of the stalk region due to the aforementioned issue of small size of the mobile region.

We would like to draw the reviewer’s attention to the modified Supplementary Figure 14, in which we compare the structure of VpOmpM1 to the full-length structure of SlpA (Farci et al. 2023). VpOmpM1 is clearly much smaller than SlpA.

Regarding the pseudosymmetry, we would like to highlight the stalk end projection plots from the molecular dynamics simulations in Supplementary Figure 3a. These plots show that the stalk trajectory has a slight bias off the C3 axis, which corresponds to the (0,0) coordinate. Therefore, we believe that the pseudosymmetry is real.

-Lines 174-191, These considerations may be made on a wrong assumption (local refinement strategies must be applied). Is PG expected to bind proteins covalently? If yes, did you use some enzyme to cut the covalent bond allowing a rational isolation of the protein? If not, how can you explain a static and stabilizing function in the presence of non covalent interactions?

In general, I would shorten this part; what makes a reference is what you observe in

VpOmpM1 isolated from *V. parvula* (its own organisms) and there the stalk seems straight and not banded (Fig. 1e).

It is known from previous work that the SLH domain of the *Selenomonas ruminantium* (Negativicutes) OmpM homologue Mep45 binds PG non-covalently (PMID: 20851903), and we have confirmed that this is the case in *V. parvula* with an in vitro binding assay in our revised manuscript (Figure 3a). We believe the OM stabilizing function is achieved by having a high copy number of OmpM proteins that form non-covalent interactions with the PG: *V. parvula* OM proteomics showed that OmpM proteins are very abundant (rank #1 for OmpM1 and #3 for OmpM2; PMID: 28713344). Although we do not know the affinity of the SLH domain-PG interaction, our qualitative PG binding experiments suggest that the interaction is reasonably strong as only minor amounts of OmpM are found in the wash fraction (Fig. 3e).

We note that the stalk in the structure of VpOmpM1 isolated from the native organism is also tilted, but not in the same orientation as the structure of the recombinant protein (Figure 1e-h).

- Line 229, the second paragraph of results is an extended mix of results and discussion. I would shorten it by focusing as much as possible on the results.

This paragraph describes Figure 3. At less than 20 lines, we do not think it is overly long and we would rather keep the few discussion sentences in order not to fragment the text too much.

- Lines 238-240, for comparison of SLH domains, you can also use the SDBC from *D. radiodurans* recently published. I am saying this because you also compared the OM region of your porin with the SDBC one (although using an incomplete structure).

Using the latest SDBC structure (published in Jan 2023) would benefit the manuscript because there you can find the complete model, including the entire stalk and SLH domain, therefore allowing you to make better comparisons.

We respectfully disagree. The N-terminal portion of the stalk region of the Farci et al. 2023 SlpA structure was built as a C α trace into a 5.3 Å map (EMD-15384). We do not believe the quality of the SLH domain in this SlpA structure is sufficient, given that there are high resolution crystal structures of SLH domains available (3PYW – 1.8 Å, 6CWH – 2 Å).

Lines 265-271, this is pertinence of the discussion section.

We have shortened this paragraph, but we believe it is best suited in the results section.

Line 295, is this a new paragraph? If yes, a title is missing. If not, it is not clear the connection with the lines above.

Thank you for pointing this out. We have introduced a new section heading (line 328).

Lines 308-310, SLH domains are very conserved as sequence and structure. Could this mean that this is not an SLH domain but just a PG-binding domain?

The N-terminal region of OmpM shows very strong homology to SLH domains. For example, for Vpar_0555 (OmpM1 in *V. parvula* strain DSM2008) the SLH domain KEGG database motif prediction has an E-value in the order of 10^{-15} (https://www.kegg.jp/ssdb-bin/ssdb_motif?kid=vpr:Vpar_0555). Also, our analysis of the extended stalk AlphaFold2 prediction, which is consistent with the cryo-EM data, shows shared features with monoderm SLH domains (Figure 3b). Conversely, there is no bioinformatics evidence that the N-terminal region folds into a PG-binding domain (IPR018537).

To avoid confusion, we would like to note that the SLH domain certainly does bind PG, as shown by our biochemical data, but it does not have a PG-binding domain fold.

Lines 318-326, again, this is pertinence of the discussion section

This paragraph describes our molecular dynamics simulation results, and we would rather keep it as is.

Figure 4, do you have any evidence of this in cryo-EM data? If this conformational changes are real, you should see the particles fall in different classes because the stalk compression is quite important. I am asking because in crystallography you might see “artifacts” due to the packing conditions and AlphaFold can only predict up to the knowledge it can find and at the moment might be too poor as there is only one published structure of a porin-like protein with a complete stalk and SLH (that in this manuscript is not even cited...)

We do not see any evidence for the compact conformation in the cryo-EM data. However, as we state in the manuscript (lines 382-383), we believe that the two conformations could be determined during biogenesis. It is possible that the crystal structure could be an artefact of crystal packing. However, in light of our new binding data (Figures 3a and 4f) we don't think this is the case—the stalk without the barrel domain cannot bind to sacculi, suggesting that the isolated stalk indeed has a different conformation, presumably observed in our crystal structure. Furthermore, it would be an implausible coincidence that the crystal structure is artefactual and AlphaFold2 predicts the same artefactual structure.

Discussion section

- Line 464, I am not sure you can state so strongly the tethering function. In this respect, it seems to me that in vivo or in situ direct evidences are missing. Apart those indirectly inferred, what are these evidences? Can you shortly cite them? It might be of help to do comparative experiments with WT strains and Mutants carrying OmpM without stalk. In this type of experiments, cryo-ET on intact cells may provide straight evidences proving or disproving your statements about tethering. I would expect from these experiments that something is loosen in the membranes. Alternatively, I would “tone down” this statement.

Thank you for this comment. Our current manuscript is following up on previous work by Witwinowski et al. 2022 (ref. 16), in which 3D-SIM and cryo-ET were used to show OM detachment in the absence of OmpM paralogues, which can be restored by expressing OmpM1 from a plasmid. In addition, we now present direct biochemical evidence for VpOmpM1 binding to PG, as mentioned above.

- Lines 477, as on the comment for Lines 308-310, SLH domains are very conserved as sequence and structure. Could this mean that this is not an SLH domain but just a PG-binding domain?

See our answer to this comment above.

- Lines 482-484, Please, see comment for figure 4. To me it seems that if these multiple conformations would be the case, then cryo-EM would have surely revealed them. On the contrary, at the moment the cryo-EM suggests that this is unlikely. I would tone-down this statement.

We have replied to the reviewer's concerns regarding the existence of multiple conformations above. We have also complied with the request to tone down the statement (lines 542-543).

- Lines 489-490, the SLH associated to the stalk was described and modeled previously on the SDBC in *D. radiodurans* (Jan 2023). Please, refer to this articles if somehow it was overlooked.

We have rephrased this sentence to include all OmpM-like porins (line 551). We note that neither the Farci et al. 2023 study, nor the von Kuegelgen et al. 2022 AlphaFold2 analysis of the SLH domain highlights the main difference between the stalked porin SLH domains and well-characterised monoderm SLH domains: the presence of three identical PG/SCWP binding sites versus non-identical sites in proteins that have three copies of the SLH domain encoded in tandem. We make this observation in this paragraph (lines 551-557). Therefore, we believe it would not be appropriate to reference these two previous studies here.

- Lines 511-512, The trimer of a β -sandwich protein (DR_0644) was observed around the stalk and in close proximity to the β -barrel interface in one SDBC structure. The DR_0644 protein was modeled in the last complete SDBC structure recently published and not mentioned in this manuscript (Jan 2023). The DR_0644 was found to be a superoxide dismutase (SOD), with several important biological implications. To me it seems that reference 29 is not the best option (it only shows an AlphaFold prediction and not functional/biological info).

Thank you for this comment. We have updated the references and changed the text to say that DR_0644 is a superoxide dismutase.

Minor comments

Introduction section

- Lines 45-46 is written ".....in presence of an additional, outer membrane", it seems that there is a second outer membrane. Maybe rephrase it as ".....is the presence of

a second external membrane, the Outer Membrane (OM).”

We have removed “additional.”

- Line 47 should be “lipopolysaccharides”

In this case, “lipopolysaccharide” is an uncountable noun as we refer to the wide variety of lipopolysaccharides found in diderm bacteria.

- Lines 51-53 I would rephrase this part. The mechanical stabilization of the OM on the cell is mainly possible because of the interaction with the PG. The PG does not provide additional stability but is (especially in the absence of an S-layer) a main static element of stability.

We refer the reviewer to ref. 4, where Huang and colleagues showed that integrity of the OM is an important element of mechanical stability for the cell. Thus, our statement that both the OM itself and OM-PG tethering is important for mechanical stability and cell morphology is accurate.

- On Lines 61 and 67, the term striking/ly to me seems redundant please use a synonym in one of the places.

The text has been rewritten to avoid repetition.

- Line 74, maybe “or” should be replaced with “and”?

This paragraph has been re-written.

- Lines 81-85, I would split this sentence in two parts

This paragraph has been re-written.

- Line 97, nutrients

Corrected.

- Lines 99-100, I would write “a OmpA-like homologue (FNLLGLLA_00518) and a porin (FNLLGLLA_00833)”

Done as requested.

- Line 194, “constructs” do you mean structures?

We have now clarified that we meant “protein constructs” (line 206).

Results section

- Figure 1, Please respect the letter order, maybe you could merge (b) with (d)?

We have swapped the labels for panels c and d and have amended the text accordingly.

- Figure 1f (caption), Please specify that you are referring to the stalk region

Done as requested.

Discussion section

- Lines 470-473, Is it known the distance between OM and IM in this species? Can the SLH domain be interacting with the inner membrane?

The distance between the OM and IM in *V. parvula* is roughly 250-350 Å, as estimated from previous cryo-ET work (ref. 16). Our new PG binding data conclusively show that the OmpM1 SLH domain binds to *V. parvula* sacculi.

- Lines 473-480, Is this species carrying an S-layer? If yes, could it be also that the S-layer could help in providing rigidity making the system less disordered?

An S-layer has never been observed in *V. parvula*. Also, even if there was an S-layer, it would be located on the outside of the cell and would not contact the periplasmic domain of OmpM.

- Lines 498-499, another DR_2577 (SlpA) structure (into its complex, the SDBC) is currently available.

We have updated the references.

- Line 503, supplementary figure 12. Please, use the complete structure available for SDBC (Jan 2023), not the truncated version, at lower resolution and with poor integrity (e.g., no cofactors bound, no intact stalk, no SLH domain....)

We have complied with this request, as stated above.

Material and methods section

- Line 660, at which temperature the centrifugation? Why such a long time?

We have specified that the centrifugation was carried out at 4°C in the revised text (line 721). The long centrifugation time was required to ensure that all cellular debris was pelleted.

Reviewers' Comments:

Reviewer #1:

Remarks to the Author:

The authors replied positively to all comments raised by me and the other reviewer and I think now the paper improved significantly. I have no other comments.

Reviewer #2:

Remarks to the Author:

The authors have addressed all comments and made extensive revisions where necessary. The manuscript has notably improved, particularly in precision within the results section. Additional experimental data have been included, specifically to illustrate the interaction between the SLH domain and PG.

Minor suggestions: 1) enhance the closing part of the abstract; 2) references 32, 37 and 71 are the same.

I am pleased with the revisions.

Reviewer #1 (Remarks to the Author):

The authors replied positively to all comments raised by me and the other reviewer and I think now the paper improved significantly. I have no other comments.

Response:

We thank the reviewer for their time and for helping to improve the manuscript.

Reviewer #2 (Remarks to the Author):

The authors have addressed all comments and made extensive revisions where necessary. The manuscript has notably improved, particularly in precision within the results section. Additional experimental data have been included, specifically to illustrate the interaction between the SLH domain and PG.

Minor suggestions: 1) enhance the closing part of the abstract; 2) references 32, 37 and 71 are the same.

I am pleased with the revisions.

Response:

We thank the reviewer for their time and for helping to improve the manuscript. As suggested, we have rephrased the closing part of the abstract and amended the references.